# Volume expansion and TRPV4 activation regulate stem cell fate in three-dimensional microenvironments

Hong-pyo Lee[1], Ryan Stowers[1] & Ovijit Chaudhuri [1]

For mesenchymal stem cells (MSCs) cultured in three dimensional matrices, matrix remodeling is associated with enhanced osteogenic differentiation. However, the mechanism linking matrix remodeling in 3D to osteogenesis of MSCs remains unclear. Here, we find that MSCs in viscoelastic hydrogels exhibit volume expansion during cell spreading, and greater volume expansion is associated with enhanced osteogenesis. Restriction of expansion by either hydrogels with slow stress relaxation or increased osmotic pressure diminishes osteogenesis, independent of cell morphology. Conversely, induced expansion by hypoosmotic pressure accelerates osteogenesis. Volume expansion is mediated by activation of TRPV4 ion channels, and reciprocal feedback between TRPV4 activation and volume expansion controls nuclear localization of RUNX2, but not YAP, to promote osteogenesis. This work demonstrates the role of cell volume in regulating cell fate in 3D culture, and identifies TRPV4 as a molecular sensor of matrix viscoelasticity that regulates osteogenic differentiation.

[1] Department of Mechanical Engineering, Stanford University, Stanford, CA 94305, USA. These authors contributed equally: Hong-pyo Lee, Ryan Stowers Correspondence and requests for materials should be addressed to O.C. (email: chaudhuri@stanford.edu)

The mechanical properties of the extracellular matrix (ECM), including ECM elasticity and stress relaxation, are key regulators of stem cell fate and behaviors, both on two-dimensional (2D) substrates[1,2] and in three-dimensional matrices[3,4]. In 2D culture, hydrogels with elasticity similar to fat (soft, ~1 kPa) or pre-mineralized bone (stiff, ~30 kPa) promote MSCs to undergo adipogenic or osteogenic differentiation, respectively[5–7]. In vivo, MSCs differentiate into osteoblasts on the 2D surfaces of osteoclast-resorbed bone in order to deposit new bone[8,9]. However, in 3D culture of MSCs in hydrogels, elasticity alone is not sufficient to determine lineage specification. In addition to elasticity, matrix remodeling significantly enhances osteogenic differentiation, and can occur through either protease-mediated degradation[10] or physical remodeling of matrices that are viscoelastic and exhibit fast stress relaxation[11]. Fracture hematomas, where osteogenic differentiation of MSCs occurs in vivo, display fast stress relaxation[11–13]. Further, understanding of the contributions of matrix viscoelasticity is relevant to the design of tissue-engineered constructs involving the culture of MSCs in hydrogels.

While mechanisms underlying mechanotransduction in 2D culture are increasingly well understood, those mediating mechanotransduction in 3D culture are less clear. On 2D substrates, cells sense and respond to stiffness by binding to ligands in ECM with integrins and generating force on the substrates via actomyosin contractility[2]. Force generation on rigid substrates promotes talin unfolding and activates vinculin[14], induces focal adhesion assembly[15] through mechanically activated focal adhesion kinase[16] and RhoA activity[17], and alters lamin A expression[6]. MSCs on stiff substrates accumulate YAP in their nuclei, and require YAP for osteogenic differentiation[18]. In 3D culture in hydrogels, osteogenesis has been found to be decoupled from cell morphology, and has been associated with integrin clustering, in physically remodelable hydrogels, and exertion of traction forces through integrins, in degradable hydrogels[3,10,11]. However, the mechanism underlying the need for matrix remodeling in 3D to induce osteogenesis of MSCs is unknown. One possibility is that matrix remodeling is required to facilitate cellular volume changes. Recently, cell volume changes on 2D substrates were determined to be significantly associated with changes in elasticity, cell morphology, and stem cell fate[19]. Further, it was found that cell volume expansion in 3D microenvironments was a key regulator of chondrocyte function[20]. These studies suggest that cell volume regulation could play an important role in dictating stem cell fate in 3D microenvironments, though the extent of volume change, effect on differentiation, and mechanism by which it might occur are all unexplored.

Here, we examine the role of cell volume in regulating MSC differentiation in 3D culture. We find that cells undergo volume expansion in hydrogels with fast stress relaxation, and that expansion is associated with cell spreading and osteogenic differentiation. Osteogenic differentiation of MSCs is reciprocally regulated by both volume expansion and activation of TRPV4 ion channels. Osteogenesis is inhibited when volume expansion is restricted, even in cells with spread morphologies. Volume expansion-mediated osteogenic differentiation is driven by increased nuclear translocation of RUNX2, but not YAP. Together, these results reveal how matrix mechanical properties regulate cell fate by enabling or restricting cell volume expansion.

## Results

**Stress relaxation promotes volume expansion and osteogenesis.**
To assess the role of cell volume expansion in osteogenic differentiation, MSCs were cultured in alginate hydrogels. Hydrogels were formed that had an initial elastic modulus of ~20 kPa, as this modulus was found previously to optimally promote osteogenesis[3] (Supplementary Fig. 1a–c). Different average molecular weights of the alginate (280 kDa, 70 kDa, and 35 kDa) were used in order to form alginate hydrogels with a range of viscoelastic responses[11]. Viscoelasticity of the hydrogels was quantified with stress relaxation tests, in which a constant strain is applied to a hydrogel and the resulting stress is measured over time. Alginate hydrogels with lower molecular weights exhibited faster stress relaxation, and calcium cross-linking concentration was adjusted to hold the initial elastic modulus constant. Prior work has demonstrated that the faster stress relaxation in the alginate hydrogels corresponds to greater creep, higher loss moduli, and higher loss tangents[20]. Degradation of the hydrogels in culture is negligible over a timescale of 7–14 days[20]. RGD, an integrin-binding motif, was coupled to the alginate to promote cell adhesion, and the final RGD concentration of the alginate hydrogels was either 1500 μM (high) or 150 μM (low).

MSCs were encapsulated in the alginate hydrogels and cultured for 7 days in equal parts of osteogenic and adipogenic induction media (mixed medium). Cellular volume increased significantly over 7 days of culture in faster relaxing gels with higher RGD density, while volume did not significantly increase in slower relaxing hydrogels (Fig. 1a, b). A general method for 3D cell volume measurement was applied to reconstruct 3D cell volume from a 2D stack of confocal microscope images[19,21,22]. Although analysis of cell volume in 3D has been shown to be prone to error[22], the accuracy of volume measurements was found to be similar across all hydrogel types (Supplementary Fig. 2a–c) and the same trend held up when image processing threshold levels were adjusted (Supplementary Fig. 2d–f). Further, the volume measurements were confirmed using super-resolution microscopy, in which such volume measurement errors are diminished (Supplementary Fig. 2g and h). In addition to volume expansion, MSCs in faster relaxing hydrogels exhibited a more spread morphology after 7 days in culture compared with cells in slow relaxing hydrogels, which were not significantly different from rounded cells immediately after encapsulation (Fig. 1c). Spread cells exhibited larger volumes than rounded cells in fast relaxing hydrogels, suggesting that cells might spread and expand their volume simultaneously (Fig. 1d). To test this possibility, both spreading and volume expansion of single cells were monitored with time-lapse confocal microscopy. Cell volume increased during the protrusion process and protruding cells had larger volumes, compared with non-protruding cells over the same observation time (Fig. 1e, f). To assess whether cell spreading was associated with cell volume expansion, MSCs were cultured in viscoelastic hydrogels with lower RGD ligand density (150 μM), which do not promote extensive cell spreading. Both volume expansion and spreading in fast-relaxing hydrogels were significantly decreased with low RGD ligand density compared with high RGD density, while significant volume expansion in slow-relaxing hydrogels was not observed, regardless of RGD density (Fig. 1g, h; Supplementary Fig. 3a–c). Prior work has demonstrated that faster stress relaxation in hydrogels promotes enhanced osteogenesis of MSCs[11]. This result was confirmed with a calculation of the percentage of cells stained positive for alkaline phosphatase (ALP) and level of osteocalcin expression (Fig. 1i; Supplementary Fig. 3d, e, 4). A significant correlation between ALP levels and cell volume was observed, suggesting the intriguing possibility that cell volume expansion may promote osteogenic commitment of MSCs (Fig.1j).

**Cell volume regulates osteogenic commitment of MSCs.** To determine the impact of cell volume expansion on stem cell fate, osmotic pressure was varied to independently modulate cell volume in fast-relaxing hydrogels. MSCs were encapsulated in

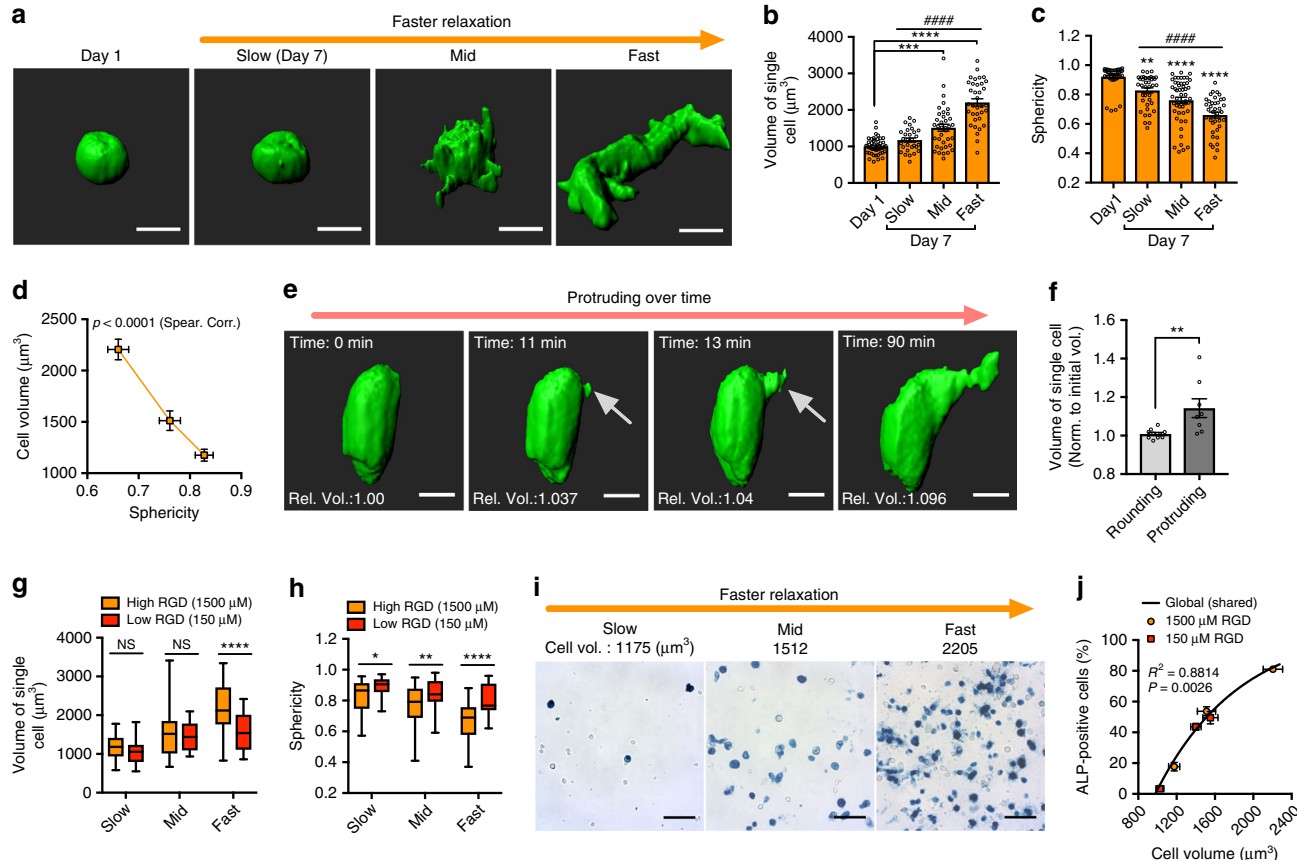

**Fig. 1** Stress relaxation regulates MSC volume expansion cell and osteogenesis. **a** Representative 3D renderings of single MSCs from confocal imaging of a cell membrane dye at the indicated levels of stress relaxation. MSCs were cultured in hydrogels with an initial modulus of ~20 kPa and different rates of stress relaxation for 7 days. Scale bar is 10 μm. **b**, **c** Quantification of cell volumes or sphericity of MSCs cultured in indicated conditions. ($n \geq 30$ single cells from three biological replications per each condition, ####$p<0.0001$ by Spearman's rank correlation, ****$p<0.0001$, ***$p<0.001$, and **$p<0.01$ compared with cells cultured for 1 day by one-way ANOVA test in **b**, **c**). **d** Scatter plot of cell volume versus sphericity of cells cultured in hydrogels with varying stress relaxation for 7 days. **e** 3D renderings of a single cell extending a protrusion in a fast-relaxing hydrogel. Arrow indicates protrusion on cell. Scale bar is 10 μm. **f** Quantification of altered volume of protruding or control cells during constant observation time (t < 2 h, $n = 10$ single cells, **$p<0.001$ by Student's $t$ test). **g**, **h** Quantification of cell volume (**g**) and sphericity (**h**) of MSCs cultured in hydrogels with different rate of stress relaxation for 7 days with a RGD density of 150 μM and 1500 μM (*$p<0.05$, **$p<0.001$, and ****$p<0.0001$ by two-way ANOVA). The box plots show 25/50/75th percentiles and whiskers show minimum/maximum. **i** Representative images of alkaline phosphatase staining (blue), indicating early osteogenic differentiation, for MSC cultured in gels of indicated stress relaxation for 7 days. Scale bar, 25 μm. **j** Scatter plot of cell volume with the percentage of cells positively stained for alkaline phosphatase produced by MSCs cultured in hydrogels with varying stress relaxation for 7 days with a RGD density of 150 μM and 1500 μM. All data show as mean ± s.e.m. MSC mesenchymal stem cell, 3D three-dimensional

hydrogels with fast stress relaxation and high RGD density, conditions in which the greatest degree of volume expansion were found, and hyperosmotic pressure was applied by adding different concentrations of 400 Da polyethylene glycol (PEG) in the induction media, as in previous studies[19,20]. The range of osmotic pressures applied was based on the effect on limiting cell volume expansion, and did not necessarily simulate conditions that might occur physiologically. After 7 days, high osmotic pressure (>197 kPa, 75 mOsm/L) significantly impaired both cell volume expansion and spreading in the fast-relaxing hydrogels, so that both of these were similar to that of MSCs in slow-relaxing hydrogels without an increase in hyperosmotic pressure (Fig. 2a–c). Notably, under high osmotic pressure, cell volume was stable and increased slightly over 7 days in culture, but was significantly lower than cells cultured in iso-osmotic conditions (Supplementary Fig. 5). Accumulation of alginate at the cell periphery was observed in fast-relaxing hydrogels with and without hyperosmotic pressure, but this accumulation was not found in slow-relaxing hydrogels (Supplementary Fig. 6). Osteogenic commitment of MSCs severely decreased under

higher hyperosmotic pressure, while CD105, a stemness marker, increased and no change was found in adipogenic markers (Fig. 2d, e, Supplementary Fig. 4, 7, and 8). Calcium, used as an ionic cross-linker in the alginate hydrogels, did not impact on osteogenic differentiation of MSCs (Supplementary Fig. 9), consistent with previous findings[3,11]. The volume of MSCs was strongly correlated to the level of osteogenic commitment of MSCs, despite distinct methods of restriction (either osmotic pressure, viscoelasticity, or RGD density). Strikingly, a single curve described the relationship between level of osteogenic differentiation of MSCs and cell volume found in all conditions (Fig. 2f).

Next, we sought to induce volume expansion with hypoosmotic pressure to further examine the regulation of osteogenesis by cell volume. MSCs were encapsulated in fast-relaxing hydrogels and cultured in induction medium for 2 days. Then, the induction medium was diluted with 20% deionized water (hypoosmotic) or an equivalent dilution in DPBS (control), and the cells were cultured for an additional 5 days (Fig. 2g). A rapid, significant increase in cell volume was observed after 1 day in hypoosmotic

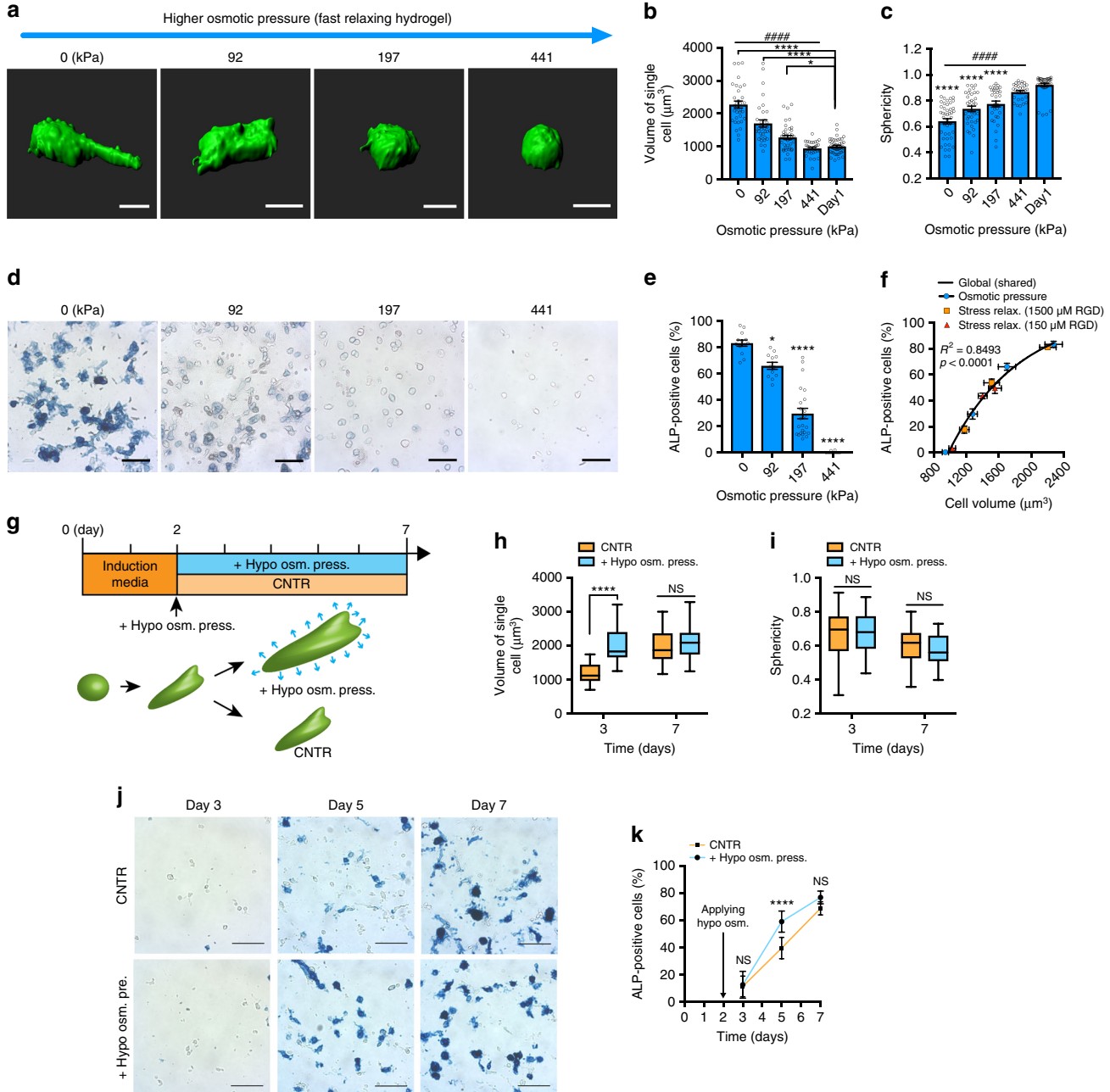

**Fig. 2** Volume expansion regulates osteogenesis of MSCs in fast-relaxing hydrogels. **a** Representative 3D renderings of single MSCs from confocal imaging of a cell membrane dye within fast-relaxing hydrogels for 7 days under the indicated osmotic pressure. Scale bar, 10 μm. **b**, **c** Quantification of cell volumes (**b**) and sphericity (**c**) of MSCs cultured in fast-relaxing gels with a RGD concentration of 1500 μM under varied osmotic pressures for 7 days ($n \geq 30$ single cells from three biological replications per each condition). **d** Representative images of alkaline phosphatase staining for MSC cultured in fast-relaxing gels under indicated osmotic pressure for 7 days. Scale bar, 25 μm. **e** Quantification of the percentage of cells positively stained for alkaline phosphatase produced by MSCs cultured for 7 days under indicated conditions ($n \geq 10$ images from three biological replications per each condition). **f** Scatter plot of cell volume with the percentage of cells positively stained for alkaline phosphatase produced by MSCs when volume of single cells was modulated by varying stress relaxation, RGD density, or osmotic pressure. A non-linear regression analysis showed a global trend of osteogenic differentiation with volume of single cells. **g** Schematic of protocol for application of hypoosmotic pressure to enhance volume expansion. **h**, **i** Quantification of cell volume (**h**) and sphericity (**i**) for MSCs cultured under hypoosmotic pressure or control conditions at days 3 and 7 ($n \geq 30$ single cells from three replications per each condition). The box plots show 25/50/75th percentiles and whiskers show minimum/maximum. **j** Representative images of ALP staining of MSCs cultured in hypoosmotic media or control media at days 3, 5, and 7 ($n \geq 10$ images from three replications per each condition). **k** Quantification of ALP-positive cells for MSCs in hypoosmotic or control medium at days 3, 5, and 7. ($n \geq 8$ images from three replications per each condition). All data are shown as mean ± s. e.m. For one-way ANOVA comparisons; *$p<0.05$, **$p<0.01$, ***$p<0.001$, and ****$p<0.0001$ and for Spearman's rank correlation; ####$p<0.0001$

pressure cultures (Fig. 2h). The final volume after 7 days was not different between the hypoosmotic group and the control, indicating that hypoosmotic conditions allow for more rapid volume expansion, but not a greater magnitude of expansion. Cell morphology was not different between the groups at any time point (Fig. 2i). Interestingly, a significant increase in ALP staining was observed at day 5, 3 days after the application of hypoosmotic pressure, indicating that the earlier increase in cell volume accelerated osteogenesis (Fig. 2j, k). As with volume expansion, no differences in osteogenesis were observed at day 7. Together, these results establish that osteogenic commitment of MSCs in viscoelastic hydrogels is regulated by cell volume.

**TRPV4 and volume expansion reciprocally control osteogenesis.** After finding the impact of cell volume on stem cell fate, we sought to identify how cell volume expands and how the volume expansion promotes osteogenic differentiation. Previous studies have demonstrated that the transient receptor potential vanilloid-4 (TRPV4) ion channels were directly activated by membrane stretching[23–26] or forces applied by ß1 integrin membrane receptor[27], and TRPV4 regulates cell volume through balancing osmolality of calcium ions in the cytoplasm[28,29], making it a logical candidate for study. Protein expression of TRPV4 in MSCs was evaluated for the different conditions and found to be diminished when cell volume expansion was restricted, either by slower stress relaxation or increased osmotic pressure (Fig. 3a; Supplementary Fig. 10). Additionally, TRPV4 ion channels were found in dense, punctate regions on the cell membrane when cell volume increased (Fig. 3b). To determine whether the increased expression and change in localization of TRPV4 were associated with increased activity, live-cell calcium imaging was performed. Intracellular calcium concentrations increased significantly in fast-relaxing hydrogels (Fig. 3c) and decreased with osmotic pressure (Fig. 3d), consistent with a functional role for TRPV4 in cellular volume expansion. The total protein concentration per cell was not significantly different across the different stress relaxation rates or osmotic pressures, indicating that the increase in cell volume is attributable to an influx of ions and water, and not accumulation of protein (Supplementary Fig. 11). It should be noted that calcium imaging measures only the relative difference in calcium concentration at a point in time, and does not reflect the cumulative change of all ions over the course of the experiments.

To evaluate the role of TRPV4 function on fate commitment of MSCs and cell volume, TRPV4 ion channel activity was modulated using a small molecule agonist or antagonist. First, TRPV4 was deactivated with GSK205, a small molecule antagonist[30,31]. Osteogenic differentiation of MSCs in fast-relaxing gels was significantly diminished by treatment with the TRPV4 antagonist compared with control treatments (Fig. 4a, b). Cell volume expansion and spreading were also significantly restricted by treatment with the TRPV4 antagonist (Fig. 4c, d). To further elucidate the role of TRPV4 ion channels in volume regulation and fate decisions, GSK101, a small molecule agonist, was used to activate TRPV4 channels[23,32]. Treatment with the TRPV4 agonist enhanced osteogenic differentiation of MSCs in hydrogels at an intermediate osmotic pressure or with an intermediate stress relaxation up to similar levels as MSCs in fast-relaxing hydrogel (Fig. 4e, f). However, osteogenic differentiation in slow-relaxing hydrogels or fast-relaxing hydrogels under high osmotic pressure was not significantly increased, despite treatment with TRPV4 agonist (Supplementary Fig. 12a, d). TRPV4 activation by the agonist also led to increased volume of cells cultured in intermediate levels of stress relaxation or osmotic pressure (Fig. 4g, h), whereas the volume and sphericity

of cells cultured in slow-relaxing hydrogels and high osmotic pressure were not altered by TRPV4 activation (Supplementary Fig. 12b, c, e, f). Interestingly, increased proliferation was observed in fast-relaxing hydrogels, low hyperosmotic pressure and TRPV4 activation (Supplementary Fig. 13a–e). However, MSCs seeded at an order of magnitude higher cell density in slow-relaxing hydrogels or fast-relaxing hydrogels under increased hyperosmotic pressure did not undergo enhanced osteogenesis (Supplementary Fig. 13f, g). Thus, it is unlikely that increases in cell density due to enhanced proliferation were the driving factor in promoting increased osteogenesis. Further, inhibition of proliferation by mitomycin C treatment did not reduce osteogenesis, and TRPV4 activation combined with mitomycin C treatment significantly enhanced osteogenic commitment in hydrogels with intermediate stress relaxation, demonstrating that TRPV4 signaling is not acting through proliferation pathways (Supplementary Fig. 13h). The TRPV4 inhibition and activation results indicate the existence of a reciprocal feedback loop between TRPV4 activation and volume expansion, which leads to enhanced osteogenic differentiation. The relationship between osteogenic differentiation and cell volume after either TRPV4 activation or inhibition fit on the previously established curve obtained in Fig. 2f (Fig. 4i), strongly supporting the conclusion that cell volume controls osteogenic differentiation. Together, these results demonstrate that osteogenic commitment of MSCs is regulated by reciprocal feedback between cell volume expansion and TRPV4 activation.

**Restriction of expansion limits osteogenesis in spread cells.** As volume expansion was associated with more spread morphologies, we next explored the role of cell morphology in mediating differentiation. Treatment with the TRPV4 agonist did not alter cell morphologies at an intermediate osmotic pressure in fast-relaxing gels or in hydrogels with medium stress relaxation, despite inducing an increase in cell volume and osteogenic differentiation (Fig. 5a, b). This observation suggests that cell morphology is independent of cell fate in 3D microenvironments. To more directly test whether volume expansion and cell fate commitment were independent of cell morphology, MSCs were cultured in fast-relaxing hydrogels with growth media, not induction media, for 7 days to permit spreading. Then induction media and hyperosmotic pressure were applied to MSCs for an additional 7 days to modulate cell volume independent of morphology (Fig. 5c, d). There were no statistically significant differences in the sphericity and surface area of cells cultured with and without osmotic pressure (Fig. 5e, f; Supplementary Fig. 14a). However, the volume of cells cultured with additional osmotic pressure was significantly reduced, indicating that cell volume expansion was severely restricted without changing morphology substantially (Fig. 5g; Supplementary Fig. 14b). Despite the maintenance in cell morphology, osteogenic commitment of MSCs significantly decreased when volume expansion was restricted by increased osmotic pressure (Fig. 4h; Supplementary Fig. 14c). This demonstrates that restricted volume expansion of MSCs controls cell fate commitment, independent of cell morphology.

**Expansion induces nuclear translocation of RUNX2 not YAP.** We next investigated the regulation of osteogenic commitment of MSCs through volume expansion at the transcriptional level. Localization of the YAP transcriptional co-regulator was first examined, as nuclear translocation of YAP has been associated with osteogenesis of stem cells in response to increased stiffness in 2D culture[5,18,33]. While there was an increase in the ratio of nuclear to cytoplasmic YAP with faster stress relaxation,

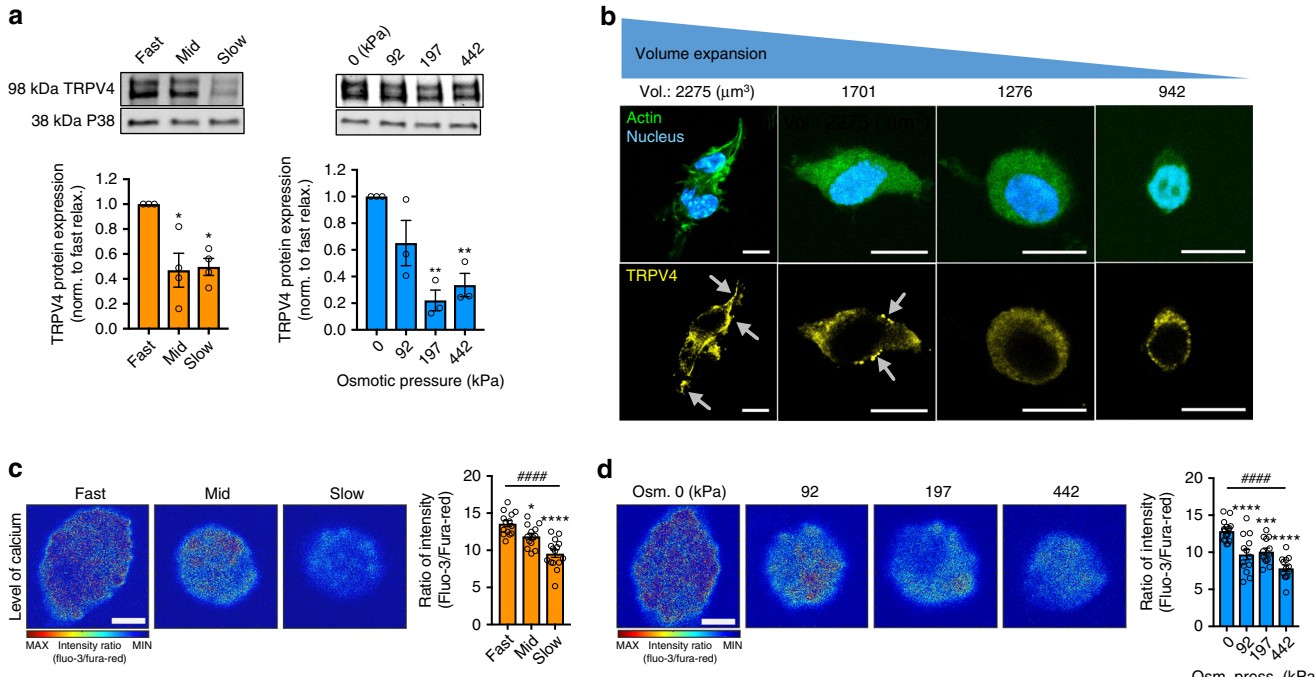

**Fig. 3** TRPV4 expression and activation is strongly correlated with volume expansion. **a** Western blot analysis and quantification of TRPV4 ion-channel protein expression (TRPV4) in MSCs cultured in hydrogel with varying stress relaxation and in fast-relaxing hydrogel with altered osmotic pressure for 7 days (**$p < 0.01$ and *$p < 0.05$, respectively, compared with fast stress relaxation and 0 kPa osmotic pressure case by one-way ANOVA test). **b** Representative images of immunohistochemical stainings of TRPV4 (yellow), actin (green), and nucleus (blue) of MSCs with different average volume of single cells. Arrows indicate a region of dense TRPV4 staining at cell periphery. Scale bar, 10 μm. **c, d** Representative images and quantification of intracellular calcium in MSCs cultured in hydrogel with varying stress relaxation (**c**) and in fast-relaxing hydrogel with altered osmotic pressure (**d**) for 7 days (****$p < 0.0001$, ***$p < 0.001$, and *$p < 0.05$, respectively, compared with fast stress relaxation and 0 kPa osmotic pressure case by one-way ANOVA test, ####$p < 0.0001$ by Spearman's rank correlation). Scale bar, 5 μm. All data are shown as mean ± s.e.m.

consistent with a previous study[11], increased osmotic pressure had no impact on YAP nuclear localization despite diminishing cell volume and osteogenesis (Fig. 6a, b). This suggests that other pathways related to volume expansion must be involved to fully drive osteogenic differentiation of MSCs. In contrast to YAP, RUNX2, a prominent transcription factor that promotes expression of osteogenic genes when translocated into the nucleus of stem cells[5,34,35], exhibited higher nuclear localization in MSCs under conditions that promote volume expansion and osteogenesis (Fig. 6c, d). Similarly, there was a significant increase in the ratio of nuclear-to-cytoplasmic RUNX2, 1 day after hypoosmotic induction (Fig. 6e). Further, exposure to the TRPV4 antagonist, which impedes cell volume expansion, suppressed nuclear translocation of RUNX2, whereas the TRPV4 agonist increased nuclear localization of RUNX2 (Fig. 6f, g). Nuclear localization of RUNX2 was highly correlated with cell volume, whereas nuclear localization of YAP did not significantly change with cell volume (Fig. 6h).

After determining RUNX2 to be mechanosensitive, we explored the upstream regulators of RUNX2 activation. RUNX2 activation is known to occur through the ERK pathway[36,37]. Pharmacological inhibition of ERK caused a significant reduction in nuclear RUNX2 and subsequent osteogenic commitment (Fig. 6i, j). However, ERK inhibition did not diminish cellular volume expansion (Fig. 6k), indicating that RUNX2 activation and osteogenesis are downstream of volume expansion. While nuclear-binding activity of RUNX2 is impacted by nuclear lamin A/C levels[38], protein levels of lamin A/C level were not found to be correlated with cell volume (Supplementary Fig. 15a, b). In addition, the role of actomyosin contractility and actin

polymerization in volume expansion and osteogenesis was investigated. In the presence of small molecule inhibitors of myosin light-chain kinase (ML-7, 25 μM), non-muscle myosin II (blebbistatin, 50 μM), or actin polymerization (cytochalasin D, 1 μM), both cell volume and osteogenesis were significantly reduced (Supplementary Fig. 16a, b). Interestingly, induced TRPV4 activity through treatment with GSK101 restored both cell volume and osteogenic commitment of blebbistatin-treated cells (Supplementary Fig. 17a, b). Thus, while actomyosin contractility and actin polymerization are important for volume expansion-mediated osteogenesis, they are not downstream of TRPV4 signaling. Together, these results indicate that volume expansion and TRPV4 activation in 3D microenvironments are involved in a mechanotransduction pathway that modulates MSC fate commitment through RUNX2.

## Discussion
We observed that in 3D microenvironments, MSC volume expanded during spreading and that volume expansion led to enhanced osteogenic differentiation. For all applied perturbations of cell volume, including modulation of stress relaxation, osmotic pressure, or TRPV4 activity, overall alkaline phosphatase levels followed a single function of cell volume (Fig. 7a). In a recent study, Guo et al. observed the opposite trend when culturing MSCs on 2D substrates; MSC volume was reduced via water efflux during spreading and promoted osteogenic differentiation[19]. Similarly, Bao et al. found that 3D microniches with smaller volumes promoted higher levels of osteogenesis in MSCs, though the microniches were larger than the initial cell volumes[21]. In both cases, there was no initial confinement of the

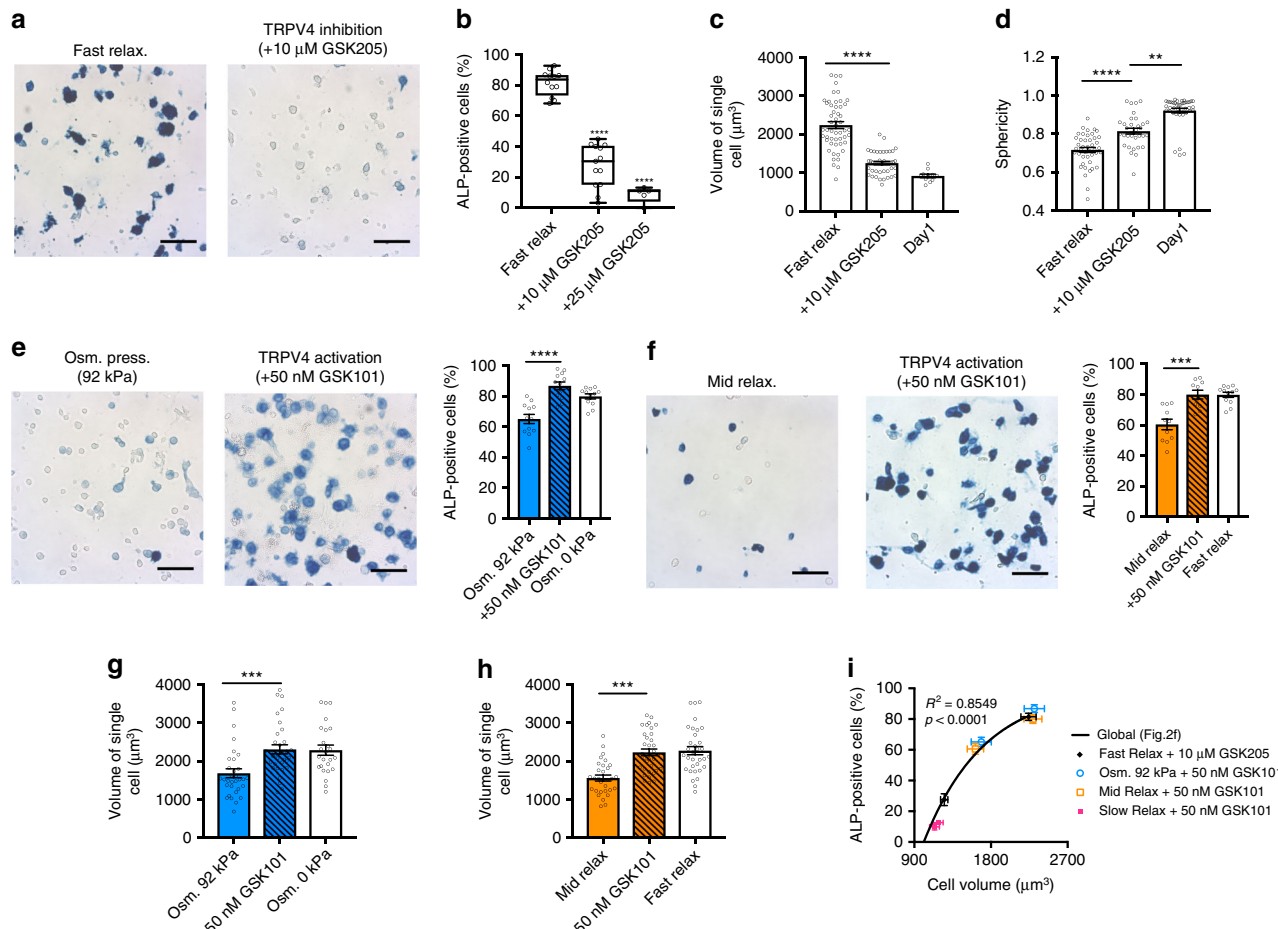

**Fig. 4** Feedback between TRPV4 activation and volume expansion regulates osteogenesis. **a** Representative images with ALP staining of MSCs cultured in fast-relaxing hydrogel with or without treatment of TRPV4 antagonist for 7 days. **b** Quantification of the percentage of cells positively stained for ALP under indicated conditions ($n \geq 10$ images from three biological replications per each condition, ****$p<0.0001$ compared with control by one-way ANOVA test). The box plots show 25/50/75th percentiles and whiskers show minimum/maximum. **c, d** Quantification of cell volumes (**c**) and sphericity (**d**) of MSCs cultured in fast-relaxing gels under indicated conditions for 7 days ($n \geq 35$ single cells from three biological replications per each condition, ****$p<0.0001$, and **$p<0.01$ by one-way ANOVA test). **e, f** Representative images of ALP staining and quantifications of the percentage of cells positively stained for ALP of MSCs cultured in fast-relaxing hydrogel under osmotic pressure of 92 kPa (**e**) and in mid relaxing hydrogel under osmotic pressure of 0 kPa (**f**) with or without treatment of TRPV4 agonist for 7 days ($n \geq 10$ images from three biological replications per each condition, ****$p<0.0001$ by student t test). **g, h** Quantification of volume of single cells cultured in fast-relaxing hydrogel under osmotic pressure of 92 kPa (**g**) and in mid relaxing hydrogel under osmotic pressure of 0 kPa (**h**) with or without treatment of TRPV4 agonist for 7 days ($n \geq 30$ single cells from three biological replications per each condition, ***$p<0.001$ by Student's t test). **i** Scatter plot of cell volume with the percentage of cells positively stained for ALP, when MSCs were cultured in indicated conditions with treatment of TRPV4 antagonist or agonist. The black line obtained by linear regression of data in Fig. 2f is shown. Scale bar, 25 μm. All data are shown as mean ± s.e.m.

cell volume expansion, so it may be that control of osteogenesis by cell volume only holds for strongly confining microenvironments. Alternatively, the culture materials were elastic and non-remodelable in these previous papers, and the geometries of the cell–substrate interfaces were flat, suggesting several other possible sources for the contrasting results. It should be noted that stiffness has been previously found as an independent regulator of stem cell fate in 3D culture, and the studies shown here were all performed using hydrogels with an initial elastic modulus of 20 kPa, as this modulus was identified as optimal for osteogenesis (Supplementary Fig. 18)[3,11]. Further, it is possible that the mechanical properties, on the scale of a cell, may vary substantially from the initial bulk values over time as the cells mechanically interact with the hydrogels, and this could play a role in mechanotransduction. Indeed, the level of alginate accumulation on the cell periphery is significantly different between hydrogels with fast and slow relaxation. However, this was

decoupled from osteogenic commitment as alginate accumulation occurred in fast-relaxing gels under hyperosmotic pressure as well, a condition in which osteogenesis levels were low. Together, our findings combined with previous studies highlight the wide range of microenvironmental features that impact MSC differentiation.

Our work shows that reciprocal feedback between activation of TRPV4 ion channels and cell volume expansion mediates osteogenic commitment of MSCs in 3D viscoelastic matrices (Fig. 7b). Previous studies demonstrated that TRPV4 channels can be mechanically activated by membrane stretching[23–25], shear stress[39,40], hypoosmotic pressure[30], and forces transmitted through integrin binding[27], and can function as a key regulator of cell volume[28,29]. Notably, TRPV4 is not strictly necessary for osteogenesis to occur, as TRPV4-null mice still develop the bone and have not reported defects in the bone[41]. We found that microenvironments that permit volume expansion of MSCs lead

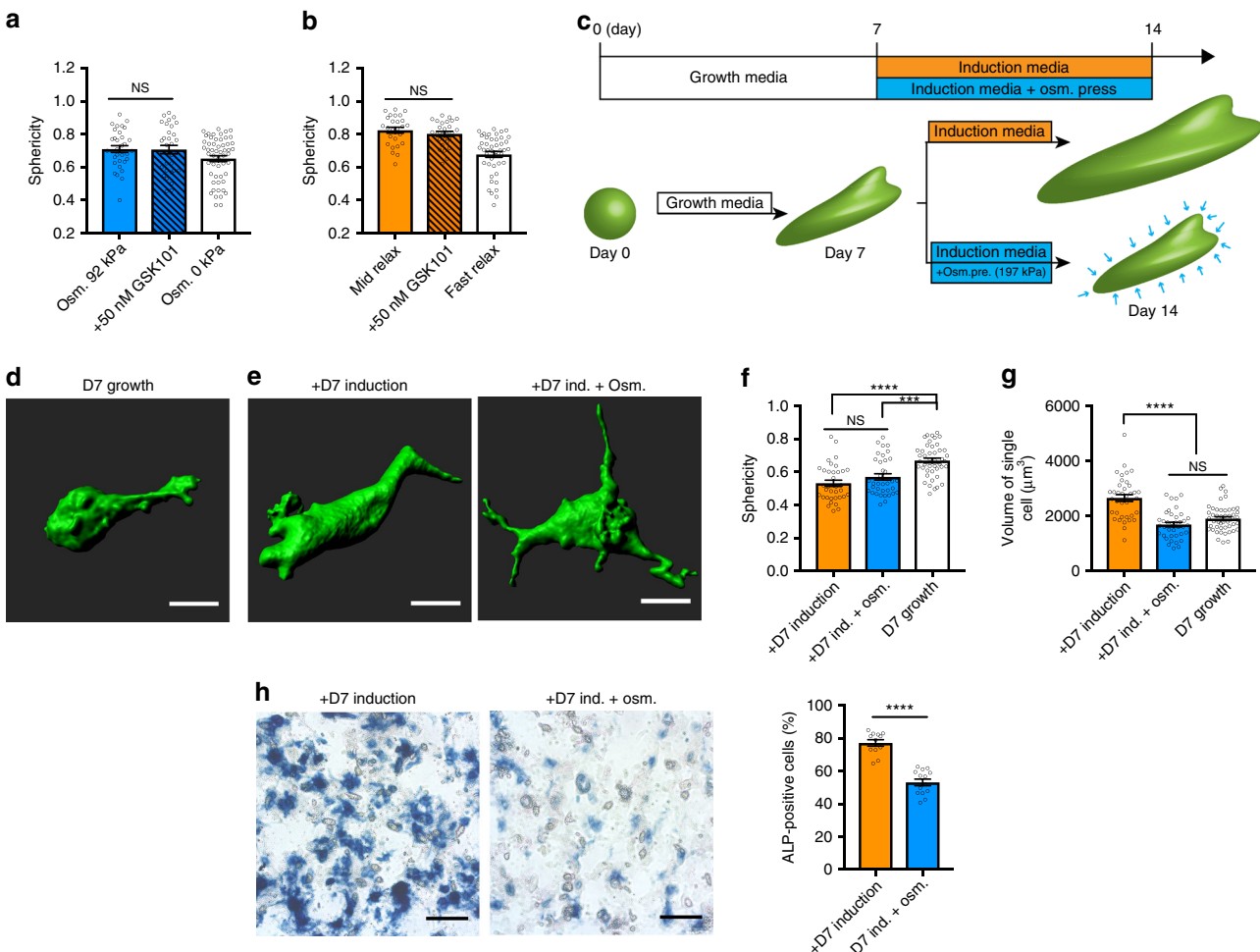

**Fig. 5** Volume expansion regulates osteogenesis of MSCs independent of morphology. **a**, **b** Quantification of sphericity of single cells cultured in fast-relaxing hydrogel under osmotic pressure of 92 kPa (**a**) and in mid relaxing hydrogel under osmotic pressure of 0 kPa (**b**) with or without treatment of TRPV4 agonist for 7 days ($n \geq 25$ single cells from three biological replications per each condition, NS $p > 0.05$ by Student's $t$ test). **c** Schematic of protocol for delayed exposure of osmotic pressure to restrict volume expansion during differentiation after culturing MSCs in growth media for 7 days. **d**, **e** Representative 3D images of a single cell cultured in fast-relaxing hydrogel with growth media for 7 days (**d**) and cultured with induction media with or without osmotic pressure for additional 7 days. **f**, **g** Quantification of sphericity (**f**) and volume (**g**) of single cells cultured in indicated conditions ($n \geq 35$ single cells from three biological replications per each condition, ****$p < 0.0001$, ***$p < 0.001$, and NS $p > 0.05$ by one-way ANOVA test). **h** Representative images of ALP staining and quantifications of the percentage of cells positively stained for ALP of MSCs cultured in indicated conditions ($n \geq 10$ images from three biological replications per each condition, ****$p < 0.0001$ by Student's $t$ test)

to increased TRPV4 expression, localization, and function. Conversely, TRPV4 activation or deactivation by small molecules regulates volume expansion or restriction, respectively. Further, volume expansion was found to be severely restricted under strong physical confinement by slow-relaxing hydrogels and hyperosmotic pressure, despite treatment with a TRPV4 agonist. These results together indicate that volume expansion and TRPV4 activation are involved in a reciprocal feedback loop that controls osteogenic differentiation. The observed dependence of volume expansion on RGD density, actomyosin contractility, and actin polymerization suggests that TRPV4 activation could be mediated through coupling to integrins and actin structures that transmit forces in addition to membrane stretching.

We observed that osteogenic differentiation via volume expansion of MSCs is independent of cell morphology in 3D culture. Previous studies have reported that cell morphology is important for fate decision of stem cells on 2D substrates[17]. However, in 3D culture, previous results have found that MSC differentiation occurs independent of cell morphology[3,10]. To assess the role of cell spreading and cell volume- in mediating

osteogenesis, MSCs were cultured in fast-relaxing hydrogels in the growth media for 7 days, prior to application of induction media and osmotic pressure. While MSCs exhibited highly spread morphologies, osteogenic differentiation was diminished when cell volume was reduced with increased osmotic pressure. Further, hypoosmotic pressure induced cell volume expansion independent of cell spreading and accelerated osteogenesis by promoting RUNX2 nuclear localization. Together, these results indicate that cell volume, and not cell morphology, controls osteogenic differentiation of MSCs in 3D culture.

The underlying mechanism by which the reciprocal feedback enhanced osteogenic commitment of MSCs involves nuclear translocation of RUNX2. Previous studies have established that RUNX2 is a major nuclear transcription factor that regulates osteogenic differentiation[34,35,42] and that activated RUNX2 is translocated to the nucleus of stem cells[5,43]. Consistent with these findings, we found a strong correlation between osteogenesis and cell volume with RUNX2 nuclear translocation. These results suggest that the reciprocal feedback between volume expansion and TRPV4 activation drives RUNX2 nuclear translocation and

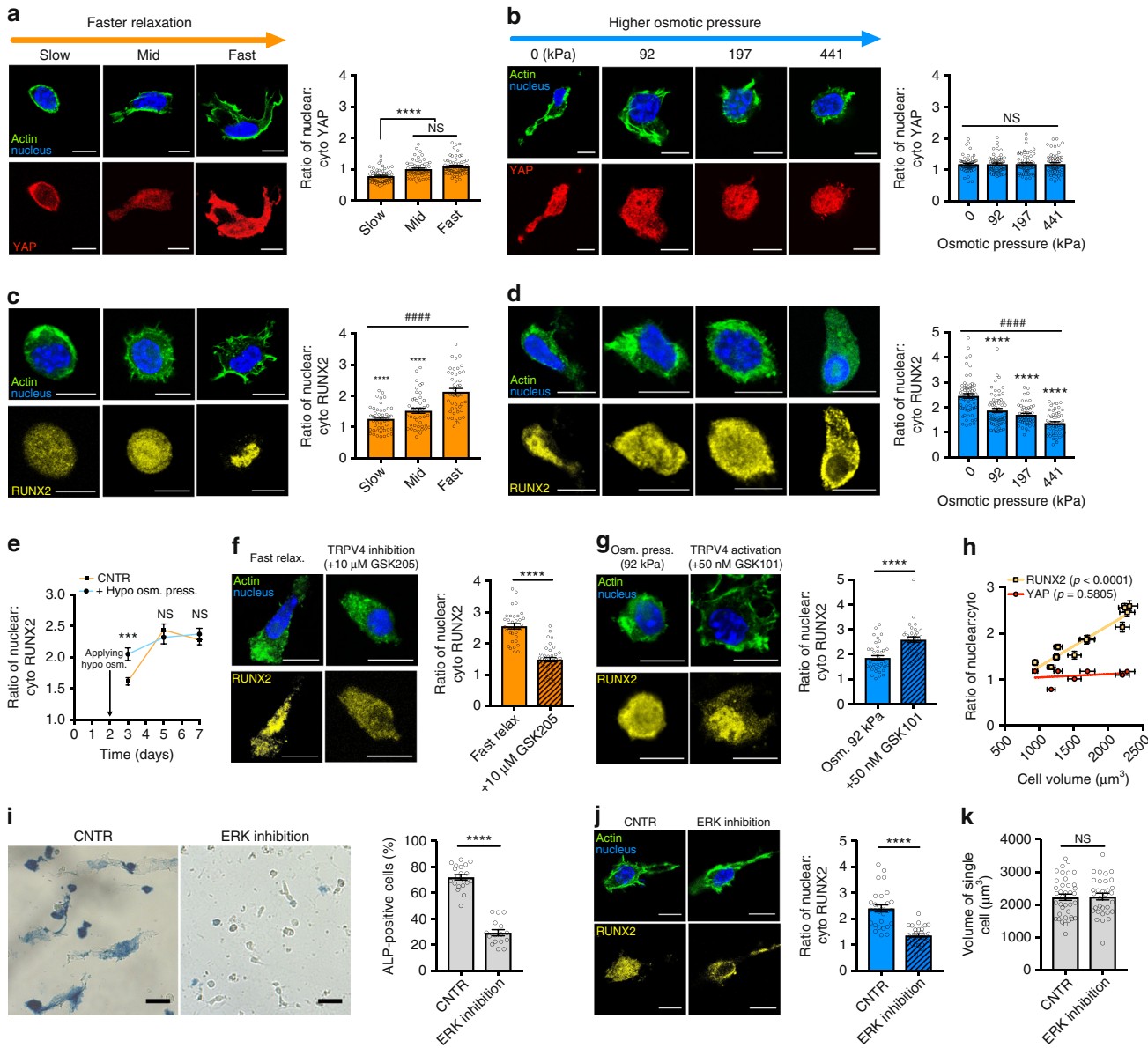

**Fig. 6** Volume expansion enhances nuclear translocation of RUNX2 but not YAP. **a**, **b** Representative images of YAP (red), actin (green), and nucleus (blue) of MSCs cultured in hydrogels with varying stress relaxation (**a**) and in fast-relaxing hydrogels with altered osmotic pressure (**b**) for 7 days. Bar graphs indicates quantification nuclear:cytoplasmic YAP ratio. **c**, **d** Representative images of RUNX2 (yellow), actin (green), and nucleus (blue) of MSCs cultured in hydrogels with varying stress relaxation (**c**) and in fast-relaxing hydrogels with altered osmotic pressure (**d**) for 7 days. Bar graphs indicates quantification of nuclear:cytoplasmic RUNX2 ratio. **e** Quantification of nuclear:cytoplasmic RUNX2 for MSCs in hypoosmotic or control medium at days 3, 5, and 7. ($n \geq 40$ images from three replications per condition). **f**, **g** Representative images of RUNX2 (yellow), actin (green), and nucleus (blue) of MSCs cultured in fast-relaxing hydrogel with or without treatment of TRPV4 antagonist (**f**) and in mid relaxing hydrogel with or without treatment of TRPV4 agonist (**g**) for 7 days. Bar graphs indicate quantification of nuclear:cytoplasmic RUNX2. **h** Scatter plot of cell volume versus nuclear:cytoplasmic RUNX2 (yellow) and YAP (red) ratios. Linear regression analysis showed a trend of nuclear RUNX2 with cell volume. Nuclear:cytoplasmic ratios were quantified with $n \geq 60$ images from three biological replications per condition. **i** Representative images of ALP staining and quantifications of ALP-positive cells for MSCs cultured in fast-relaxing hydrogels with or without treatment of the ERK inhibitor for 7 days ($n \geq 15$ images from three replications per condition). Scale bar, 25 μm. **j** Representative images of RUNX2 (yellow), actin (green), and nucleus (blue) and quantification of the nuclear:cytoplasmic RUNX2 ratio ($n \geq 30$ images from three replicates). **k** Quantification of the volume of single cells cultured in indicated conditions ($n \geq 30$ cells from three replications per condition). Scale bar in all immunostaining images is 10 μm. All data are shown as mean ± s.e.m. For one-way ANOVA comparisons and Student's $t$ test; *$p<0.05$, **$p<0.01$, ***$p<0.001$, and ****$p<0.0001$ and for Spearman's rank correlation; ####$p<0.0001$

enhanced osteogenic differentiation. Similar to previous studies[10,11], cell volume expansion in our system was mediated through myosin contractility and actin polymerization. Inhibition of ERK signaling significantly reduced RUNX2 nuclear translocation and decreased osteogenesis without a change in cell volume, implicating ERK signaling to be downstream of cell volume expansion. YAP nuclear translocation and lamin A/C expression did not correlate with cell volume and alkaline phosphatase levels in these studies, contrasting the expectation from previous 2D studies finding increased YAP activation

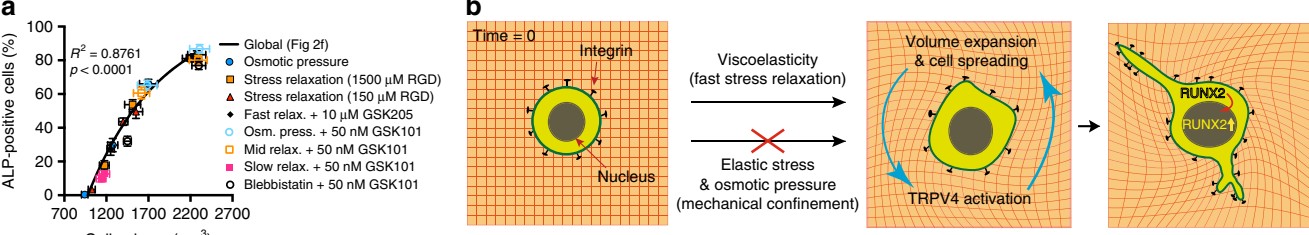

**Fig. 7** Volume expansion and TRPV4 dictate cell fate response to viscoelasticity. **a** Percentage of ALP-positive cells as a function of cell volume for all the different conditions probed in this study appear to follow a single relationship. **b** Cartoon schematic of proposed mechanism. MSCs cultured in viscoelastic matrices physically remodel their microenvironment to expand cell volume associated with spreading. The volume expansion is interdependent with TRPV4 activation, which enhances osteogenic differentiation of MSCs by increasing nuclear localization of RUNX2. In this process, mechanical confinement by either elastic stress or osmotic pressure of surrounding environment restricts cell volume expansion, impeding osteogenic commitment

associated with enhanced osteogenesis[18]. However, it may be that a basal level of nuclear YAP signaling is still necessary for osteogenic differentiation.

We demonstrated that cell spreading and volume expansion occurred simultaneously in viscoelastic microenvironments with fast stress relaxation. While a previous study observed that chondrocytes, a non-adherent cell, exhibited isometric cell volume expansion in fast-relaxing hydrogels[20], we observed that MSCs generally exhibited non-isometric expansion via integrin-mediated protrusions in high-density RGD, fast-relaxing hydrogels. Since the alginate hydrogels used here were nanoporous and non-degradable by proteases, cell spreading and volume expansion necessitate physical remodeling of the matrix. These lend further support to the notion that the ability of a 3D microenvironment to be remodeled is a critical property in regulating stem cells behavior. However, physical remodeling of the matrix in the absence of cell volume expansion, as observed for MSCs in fast-relaxing gels under high hyperosmotic pressure, was not sufficient to drive osteogenesis, highlighting the importance of cell volume expansion. While differentiation takes place over days, cell spreading and volume expansion occur continuously over a timescale of minutes to hours. This explains why altering hydrogel stress relaxation over a timescale of minutes to hours has a potent impact on differentiation. Previous studies have found that degradation of more elastic hydrogels is crucial for cell fate decision in 3D culture[10,44,45]. Our results suggest that degradation could be required to make room for cell volume expansion in such gels.

This study presents reciprocal feedback between volume expansion and TRPV4 activation as a mechanism mediating underlying cell fate determination for MSCs in 3D culture. More generally, these findings reveal cell volume expansion in confining 3D microenvironments as a key regulator of stem cell fate. This provides insight on the underlying mechanism of stem cell mechanotransduction during in vivo bone fracture healing, and offers important considerations to the design of biomaterials used in tissue engineering.

## Methods

**Alginate preparation**. The molecular weight of alginate was modified by irradiation with a cobalt source[11]. Sodium alginate (average molecular weight; 280 kDa, LF20/40, FMC Biopolymer) was irradiated with a 3 or 8 Mrad cobalt source to decrease the average of molecular weight to 70 kDa or 35 kDa, respectively. The alginate was then dialyzed in DI water, purified with activated charcoal, sterile filtered, frozen, and lyophilized. RGD peptides were coupled to alginate using carbodiimide chemistry[11]. Alginate was dissolved in 0.1 M MES buffer. N-(3-dimethylaminopropyl)-N'-ethylcarbodiimide hydrochloride (EDC, Sigma-Aldrich), Sulfo-N-hydroxysulphosuccinimide (sulfo-NHS, ThermoFisher Scientific), and GGGGRGDSP (Peptide 2.0) peptides were sequentially mixed in the alginate solution for 24 h. The final peptide concentration in the alginate gels was

either 150 μM or 1500 μM. For fluorescent alginates, fluoresceins (Acros Organics) were coupled to RGD-alginate with carbodiimide chemistry[11].

**Mechanical characterization**. The mechanical properties of alginate hydrogels were measured with compression testing[20]. Briefly, alginate hydrogels were prepared with 6-mm diameter and 2-mm thickness. The alginate hydrogels were equilibrated for 1 day in standard Dulbecco's Modified Eagles Medium (DMEM, Invitrogen) and compressed with an Instron 5848 mechanical tester to 15% strain with a deformation rate of 1 mm/min and held at 15% strain for $10^4$ s. Stress derived from deformation of the hydrogels was recorded over time. The initial modulus of the alginate was measured with the slope of stress–strain curve within 5–10% strain range. The rate of stress relaxation of each alginate was calculated as the time until the initial stress reached at 15% strain decreased to half of the original value. This time ($\tau_{1/2}$) is used as a metric for comparison, and relaxation occurs continuously.

**Encapsulation of cells within hydrogels**. D1 MSC cells (ATCC, CRL-12424) were cultured in the growth medium containing high-glucose DMEM (ThermoFisher Scientific) 10% fetal bovine serum (FBS, GE Healthcare), and 1% penicillin/streptomycin (ThermoFisher Scientific) to 70% confluency before passaging. Cells were mixed with the alginate solution prior to cross-linking to achieve a final cell concentration of 10 million/mL. Cell volume before encapsulation and 1 day after encapsulation were found to be similar (Supplementary Fig. 19). Cells were cultured in gels for either 7 days in a 50%/50% mixture of the osteogenic induction medium and adipogenic induction medium (mixed medium), or for 7 days in the growth medium followed by 7 days in the mixed medium. The osteogenic induction medium contained 50 μg/ml L-ascorbic acid (MilliporeSigma) and 10 mM β-glycerophosphate (MilliporeSigma) in the growth medium. The adipogenic induction medium contained 0.1 μM dexamethasone (MilliporeSigma) in the growth medium.

**Sample preparation for imaging**. D1 MSCs cultured in alginates hydrogels were fixed with 4% paraformaldehyde (PFA, Alfa Aesar) for 1 h, and the alginate hydrogels washed three times with PBS containing $Ca^{2+}$ and $Mg^{2+}$ (cPBS, GE Healthcare). The alginate gels were placed in 30% sucrose (Fisher Scientific) overnight and then in 30% sucrose-OCT compound solution (Tissue-Tek) for 5 h, sequentially. The alginate gel was embedded in OCT, frozen, and sectioned to 25-μm or 100-μm thickness using a cryostat (Leica CM1950). The sectioned samples with 25-μm thickness were used to stain alkaline phosphatase (ALP) or for immunohistochemistry, and the 100-μm sectioned samples were used for image analysis to measure cell volume and sphericity.

**Immunohistochemistry**. The sectioned samples were washed three times in DPBS, permeabilized with 0.5% Triton X-100 (Sigma) in DPBS for 15 mins, and blocked for 1 h with 1% bovine serum albumin (BSA, Sigma), 10% goat serum (Invitrogen), 0.3 M glycine (Fisher), and 0.1% Triton X-100 in DPBS. The samples were incubated overnight with the primary antibody [TRPV4 (Abcam, ab39260), RUNX2 (Abcam, ab76956), YAP (Santa Cruz, sc-101199), CD105 (R&D Systems, MAB1320), all used at 1:200 dilution]. After washing three times with 0.1% Triton X-100 in DPBS, DAPI and phalloidin (Invitrogen) were incubated for 1 h to stain the nucleus and F-actin along with secondary antibody, such as goat anti-Rabbit IgG AF647 (Invitrogen, 1:500 dilution) and goat anti-mouse IgG AF 555 (Invitrogen, 1:500 dilution).

**ALP staining**. ALP was stained with a FastBlue working solution of 500 μg/ml Fast Blue BB (MilliporeSigma) and 500 μg/ml naphthol-AS-MX (MilliporeSigma) phosphate in an alkaline buffer (100 mM Tris-HCl, 100 mM NaCl, 0.1% Tween-20,

50 mM MgCl2, pH = 8.2). Slides were first washed three times in DPBS, equilibrated in alkaline buffer for 15 min, then incubated in FastBlue working solution for 60 min at room temperature. The slides were then washed in alkaline buffer for 15 min followed by 15 min in DPBS. Slides were imaged using a Zeiss Axiovert 200 M microscope with a 20× objective.

**Quantitative polymerase chain reaction.** MSCs were extracted from alginate gels by incubating gels in ice cold 50 mM EDTA in DPBS for 10 min with pipette mixing. The suspensions were centrifuged at 500×g for 10 min to pellet cells. TRIzol (ThermoFisher Scientific) reagent was added, and the cells were lysed by passing through a 30-G syringe. RNA was isolated by phenol–chloroform extraction and RNA extraction columns (Green Bio). One microgram of RNA was reverse transcribed into cDNA using the High-Capacity Reverse Transcription Kit (Applied Biosystems). Fast SYBR green master mix was used, with primers listed in Supplementary Table 1. Reactions were performed on an Applied Biosystems 7500 instrument.

**Western blot.** Cell pellets were extracted from alginate gels as described above. The pellets were lysed in RIPA lysis buffer with protease and phosphatase inhibitors (ThermoFisher Scientific), and the protein concentration was determined using the BCA assay. Samples were diluted in Laemmli Sample Buffer (Bio-Rad) to 2.5 μg/μl and 25 μg was loaded in 4-15%, 15-well gels (Bio-Rad). The gels were run for 35 min and the protein was then transferred to a nitrocellulose membrane (Bio-Rad). The membrane was blocked in 5% non-fat milk for 1 h then incubated with primary antibodies overnight (TRPV4, Abcam ab39260, 1:300, p38 Santa Cruz Biotech. sc-535, 1:1000, osteocalcin, Alfa Aesar, J65216, 1:250, lamin A/C, Cell Signaling Technology, #2032, 1:500). Secondary antibodies against the primary (Licor, IRDye800CW) were incubated with the blot for 1 h. The blots were imaged using a Licor Odyssey imaging system.

**3D image analysis.** Using 100-μm sectioned samples, the cell membrane was stained with a octadecyl rhodamine B chloride (R18, ThermoFisher Scientific) and the nucleus was stained with DAPI in DPBS for 20 min. After washing three times, a Leica SP8 confocal microscope was used to take 3D image stacks with a 63X NA1.40 PlanApo oil immersion objective. Analysis of cell volume and sphericity was based on single cells not in contact the other cells. Optical image stacks of single cells were obtained with a 0.4-μm z-axis interval. For live cell images, cells were labeled with R18, washed two times with DPBS, and encapsulated in hydrogels. A 10X NA = 0.4 dry objective was used to take 3D stacks for live cell imaging with 1-μm z-axis interval and comparable pixel distance of x–y plane. Image stacks were visualized, and cell volume and sphericity were automatically calculated using Imaris software. The threshold for 3D visualization was held constant for each batch of experiments. Volume was calculated by the number of voxels. To examine the uncertainty of volume measurement with altered threshold, cell volume was quantified with an adjusted threshold increased by the standard deviation, or the optimized threshold that the Imaris program automatically determines in each 3D-image to remove user bias. Fluorescent microbeads of 1-μm or 15-μm nominal diameter were used to validate the volume measurement technique and ensure that the various hydrogels did not alter the measured volume. Airyscan, a super-resolution microscopy technique[46], was used with a Zeiss 880 confocal microscope and deconvolution processing to validate the cell volume measurements with higher resolution. Sphericity was obtained with the following formula.

$$\text{Sphericity} = \frac{\pi^{\frac{1}{3}}(6V_c)^{\frac{2}{3}}}{A_c} \quad (V_c = \text{volume of single cell, } A_c = \text{surface area of single cell})$$

(1)

Sphericity calculated from this formula yields a value of 1 for a sphere and 0 for a line.

**Two-dimensional image analysis.** YAP and RUNX2 were stained with primary antibodies listed above for analysis of transcription factor localization. DAPI and phalloidin were used to stain the nucleus and cellular domain, respectively. A Leica SP8 confocal microscope with a 63X NA1.40 PlanApo oil immersion objective was used for imaging. For all image analysis, cellular area and nuclear domain were, respectively, obtained with phalloidin image and DAPI image by applying Otsu's thresholding method. Non-nuclear cytoplasm domain was found by subtracting nuclear domain from whole cellular area. Each domain was applied as a mask to obtain the intensity of target protein in each domain. Using the obtained images and the intensities, the fraction of nuclear localization of target protein was calculated by dividing the average intensity of target protein over the nuclear area to the average intensity of the target protein over the non-nuclear cytoplasm area. The nuclear-to-cytoplasmic ratio determined from 2D images was highly correlated with and not significantly different from the same ratio determined from a 3D volumetric analysis throughout the entire cell (Supplementary Fig. 20a–d). For 3D volumetric analysis of nuclear-to-cytoplasmic ratio, the similar procedure for 2D images was conducted. Nuclear domain and non-nuclear cytoplasm domain were, respectively, obtained from each 3D stack of DAPI and phalloidin image for single cell by applying Otsu's thresholding method. The intensity of target proteins was obtained with the mask of each domain in 3D stack of target protein image. The

fraction of nuclear localization of target protein in 3D stack was calculated by dividing the average intensity of target protein over the nuclear volume to the average intensity of the target protein over the non-nuclear cytoplasm volume[45].

**Inhibition studies.** For inhibition or activation of TRPV4 ion-channel function, pharmacological antagonist (GSK205, Calbiochem) or agonist (GSK1016790A, Calbiochem) were added to the induction media for culture over 7 days, respectively. In order to inhibit ERK, myosin activity, and actin polymerization, PD98059 (10 μM, Tocris Bioscience), ML-7 (25 μM, Tocris Bioscience), blebbistatin (50 μM, Tocris Bioscience), or cytochalasin D (1 μM, Tocris Bioscience) were applied to the induction media, respectively. To inhibit proliferation of D1 MSCs, mitomycin C (20 μM, Sigma-Aldrich) was applied to cells for 1 h before trypsinization. The cells were washed three times with DPBS and then encapsulated in hydrogels. ALP analysis and volume measurement were then performed. The concentration of each small molecule was similar to concentrations used in previous studies[3,11,23,30].

**Osmotic pressure studies.** Modulation of osmotic pressure in the culture media was performed with 400 Da polyethylene glycol (PEG 400, TCI America)[19,20]. The hyperosmotic pressure of the induction media was modulated by adding 400 Da polyethylene glycol. PEG 400 concentrations of 0% wt/vol (0 mOsm/L), 1.5% wt/vol (37.5 mOsm/L), 3% wt/vol (75 mOsm/L), and 6% wt/vol (150 mOsm/L) in the induction media, respectively, applied 0 kPa, 92 kPa, 197 kPa, and 441 kPa of osmotic pressure to cells encapsulated in fast-relaxing hydrogels. Osmotic pressures were calculated with concentration of applied PEG 400 by using empirically obtained formula:[47,48]

$$y(\text{atm}) = 0.00002c^4 - 0.0007c^3 + 0.0311c^2$$
$$+ 0.5596c \,(\text{wt/vol concentration of PEG 400 in media is indicated as } c)$$

(2)

The hypoosmotic pressure of the induction medium was altered by diluting induction medium with 20% deionized water or an equivalent dilution with DPBS as control group.

**Protein-level measurement.** Cells were extracted from alginate gels with the same procedure done for western blot. The pellets were lyzed in RIPA lysis buffer with protease and phosphatase inhibitors (ThermoFisher Scientific). The lysis solution was used to measure the protein concentration by using the BCA assay and the DNA amount by using the PicoGreen assay (Molecular Probes).

**Calcium imaging.** D1 MSCs cultured in alginates hydrogels for 7 days were incubated with Fluo-3 AM (20 μM, ThermoFisher Scientific) and Fura-red AM (33 μM, ThermoFisher Scientific) for 1 h, washed three times with DPBS and incubated in the induction media before live cell imaging. To record intensity of calcium dyes, a Leica SP8 confocal microscope was used with a 20× NA0.8 objective. The Fluo-3 and the Fura-red was excited at 488 nm and the emissions were, respectively, detected at 515–580 nm (Fluo-3) and at > 610 nm (Fura-red)[49,50]. Relative calcium concentration in cells was converted from the ratio of the Fluo-3 intensity to the Fura-red intensity[51,52].

**Hydrogel accumulation measurement.** D1 MSCs were encapsulated in hydrogels containing fluorescent-tagged RGD alginates that had 40 μM FITC and 1500 μM RGD as final concentrations and different levels of stress relaxation. The cells were cultured in the hydrogels with or without hyperosmotic pressure for 7 days. Images of the fluorescent alginates nearby each cell were taken with a Leica SP8 confocal microscope with a 20× NA0.8 objective. The level of alginate accumulation was quantified as the ratio intensity of fluorescent alginate within 3 μm of the cell border to the background intensity of the hydrogel at 30 μm away from each cell[11].

**Statistical analysis.** All experiments were analyzed with biological triplicates and three replicate hydrogels per condition. Statistical comparisons were performed with GraphPad Prism 7.0 software (GraphPad Software). All analyses of cell volume and sphericity were performed with $n \geq 40$ cells per conditions. Spearman's rank correlation test was performed to assess the significance of data trends. One- or two-way ANOVA tests were applied to compare data between multiple groups with a Bonferroni post hoc comparison and a two-tailed Student's $t$ test was used to compare two experimental groups. All specific information for statistical analysis are presented in Supplementary Table 2.

**Reporting Summary.** Further information on experimental design is available in the Nature Research Reporting Summary linked to this article.

## Data availability
Data supporting the findings of this paper are available from the corresponding author upon reasonable request. A Reporting Summary for this article is available as a Supplementary Information file. The source data underlying Fig. 1b-d, f–h, j, 2b–c, e–f, h–i, k, 3a, c, 4b–i, 5a–b, f–h, 6, 7a and Supplementary Figs 1, 2b-f, h,

3b–c, e, 4a–b, 5, 6b, 7b–d, 8, 9, 11, 12a–f, 13a–g, 14a–c, 15a–b, 16a–b, 17a–b, 18, 19, and 20a–d are provided in a Source Data file.

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

## Acknowledgements

The authors acknowledge Katrina Wisdom for providing fluorescent alginate and all members of the Chaudhuri lab for helpful discussions. We also acknowledge the Stanford Cell Sciences Imaging Facility for access to the Zeiss confocal microscope and Imaris software. This work was supported by a Stanford Bio-X fellowship (to H.L.), National Institutes of Health fellowship (F32CA210431 to R.S.), and a National Institutes of Health grant (R21 AR074070 to O.C.).

## Author contributions

H.L., R.S. and O.C. designed the experiments. H.L. and R.S. conducted experiments and analyzed the data. H.L., R.S. and O.C. wrote the paper.

## Additional information

**Competing interests:** The authors declare no competing interests.

