## [Peer Review File · Nature Communications]

Reviewers' Comments:

Reviewer #1:

Remarks to the Author:

Review report (NCOMMS-18-05610)

Lee et al reported an interesting work showing volume of MSC changes in 3-dimensional hydrogels upon different compositions and regulates stem cell fate. The finding is interesting and potentially important to understand cell behavior in 3D. The paper is well written and understandable to a wide audience. Nonetheless, I'm not fully convinced based on the current submission due to the following reasons.

My first major issue is about the volume measurement in 3D. The authors should report the accuracy of cell volume determinations. It will be possible to estimate the measurement uncertainty by calculating how the error of the fluorescence threshold value propagates to the cell volume. It is not clear how the threshold is obtained in the work, and what is the uncertainty of this threshold. Currently, the authors only state that the same threshold is used in this study. This does not tell us the uncertainty of this chosen threshold. How much does the calculated cell volume change when the threshold value is increased or decreased by the standard deviation? The z resolution of the standard confocal microscope is approximately $1.4 \times \text{wavelength} \times \text{refractive index} / \text{NA}^2$ which is about 0.5 microns when respective parameters are 543nm, 1.4NA and in water. This axial resolution is the uncertainty of cell boundary localization in z-axis for an ideal case on a coverslip. In different gel compositions in 3D, the point spread function can significantly deviate from the one on the surface of a coverslip because gels can relax and swell, therefore the measurement uncertainty will be significantly affected and will also not be the same in different gel conditions. This can be clearly seen in figure 1a and 1e where the z-resolution seems ill in the spreading cell case. How much uncertainty in cell volume measurement due to this low and varying z-resolutions in this study should be examined and discussed.

The cell volume expansion observed here presumably is interesting, however, no understanding is proposed. What content is changed, cellular water or proteins?

Further, how are the cell volume change and homeostasis achieved and maintained, at different extracellular matrix conditions? The authors seem to suggest that the viscoelastic relaxation allows the cell volume to expand. If so, this indicates that elastic matrix confinement is limiting cell volume in the slow relaxation gel. If this is the case, are there any elastic interaction between the matrix and the cell? In another word, is the elastic force from the matrix involved in setting the equilibrium cell volume? Or are the osmotic balance across cell membrane always maintained at different matrix relaxation conditions? If the osmotic balance is always maintained, as typically assumed, the 2 fold volume expansion induces the total amount of internal osmolytes to double. What are the dominating osmolytes, ions, small proteins, or ATP? Are the total number important for stem cell fate? These are critical for understanding cell volume expansion and their impact.

The same questions continue in the osmotic compression condition. How stable is the cell volume under such non-isotonic condition? Again, if the osmotic balance is still maintained, the concentration of ions must increase in the cell to balance the additional osmotic pressure generated by PEG400 in the medium. If this is the case, the concentration of ions in the cell and that in the medium will not be balanced. In the meantime, various ions pumps are known to function if there is an imbalance in a particular type of ions. How is a dynamic osmotic balance still maintained in such cases?

It is claimed that volume expansion is driving the osteogenic differentiation, however, no controlled volume expansion experiments are performed. For example, the authors can increase cell volume by an alternative approach, perhaps hypotonic pressure, and show that osteogenic differentiation is still enhanced. The osmotic pressure experiment reducing cell volume is also not

convincing enough as osmotic pressure can cause many issues to cell behavior and physiology, for example, can reduce cell proliferation significantly. Understanding the mechanism of cell volume expansion/change is therefore also important here since it will allow us to know what content is being regulated during volume expansion in the fast relaxation gel, therefore helps the design of control experiments to regulate cell volume to study its impact.

Also, cell morphology changes significantly upon cell spreading, suggests that nuclear morphology could change therefore nuclear lamin expression could also change. Similarly, under osmotic compression, cell nuclear size and morphology must change, therefore, can also affect nuclear lamin expression. As nuclear lamin has been demonstrated repeatedly to play a role in stem cell fate (reference 6 in the manuscript), the status and role of nuclear lamin should be commented.

Another issue is that cell volume change is associated with cell spreading. Even under additional osmotic pressure, as the authors claim that cell spreading is not affected, what about cell surface area or other morphological factors? My question is that how could the authors be sure that the observed effect in stem cell fate is due to cell volume, not cell surface area or the number of branches? I find it unclear to conclude that cell volume expansion is the cause for stem cell fate change. What the authors found is potentially important, however, it is unclear if cell volume is the cause here, not merely a consequence.

Minor:

1. During encapsulation, what is the osmotic pressure of the solution including medium and polymers and ions? Is there any volume increase or decrease during encapsulation?

2. Clear YAP localization change under varying osmotic pressures has recently been observed by several different groups, for example in several works by Kunliang Guan's group, and another one by Alberto Elosegui-Artola et al, Cell, 2017. YAP/TAZ are known to regulate stem cell fate. How do we understand the difference in the YAP localizations in those studies and in the current one?

3. What is the role of matrix stiffness here? As reference 3 in the current manuscript showed, decreasing matrix stiffness causes less osteogenesis.

4. What exactly is the role of TRPV4 and how they are involved in volume regulation is not clear from the current text.

Reviewer #2:

Remarks to the Author:

The authors describe an apparent correlation between cell volume and efficiency of MSC osteogenic differentiation in 3D alginate culture that seems to depend on TRPV4 activation. First, MSCs were cultured in 20kPa alginate gels with varying relaxation times, where gels with fast relaxation time allows for cell spreading (increase of cell volume) and effective osteogenic differentiation. Inhibition of cell spreading by hyperosmotic challenge (by addition of PEG) seems to reduce the osteogenic differentiation efficiency of cells seeded in gels with fast relaxation time. The effects can be recapitulated by TRPV4 inhibition and rescued by TRPV4 activation. Additionally, by immunostaining for RUNX2, hyperosmotic stress and TRPV4 regulation also affect RUNX2 translocation, which is essential for osteogenic differentiation. Considering the growing reports on the role of TRPV4 in mechanotransduction and the lack of knowledge in how MSCs perceive 3D environments, this study is timely and will expand the field. The data of this study is clear and the role of TRPV4 is apparent. However a few concerns temper enthusiasm:

1. New bone is typically formed in vivo by MSC-generated osteoblasts on top of osteoclast-resorbed bone, which is a 2-dimensional process. In what precise situations is this 3D approach

relevant to the biology of bone? Further, is there reason to believe that in bone, osmolarity differs in ways studied here? And when will cell volume regulation play a role in such process? How do the mechanical parameters of these gels compare to that of osteoid?

2. How do the mechanical parameters of these gels vary in space and time around the cells?
3. Cell volume has long been known to be regulated by cytoskeleton (eg. cytochalasin) and vice versa. De-coupling should be examined. TRPV4's role in regulating cell contractility has also been widely reported, given its primary function as a calcium channel. What is the effect of TRPV4 regulation and prolonged hyperosmotic challenge on actomyosin organization and contractility.
4. Seeding in gels with slow relaxation time, hyperosmotic media and TRPV4 inhibition all seem to decrease cell count (per images), while TRPV4 activation does the opposite. Given the likely importance of cell density in osteogenic differentiation, several things need to be checked for all the conditions:
 - a. Effects of seeding density (e.g. does seeding more cells in slow relaxation time gel alter ALP production?) or are all of these single cells truly isolated?
 - b. Effects of TRPV4 regulation and hyperosmotic media on cell proliferation
5. It is not clear from the citation provided (33) what the role of nuclear translocation of RUNX2 is. This should be more fully explored here, including RUNX2's phosphorylation.
6. Fig.3: What are the contributions of TRPV4 activity vs TRPV4 expression levels (abundance, Fig.3A) vs TRPV4 localization/clustering (Fig.3B) in volume regulation, differentiation, etc.? What aspect of TRPV4 is most important here?

Minor points:

1. Fig. 1E is missing arrows that are mentioned in the figure caption.
2. Fig. 5: labels for the images are missing (e.g. blue is Nucleus)
3. Please include the conversion of kPa osmotic pressure to mOsm/kg or mOsm/L (unit of osmolality) in the method section.
4. Fig 1D: Plot is probably not needed with the data present in b & c.
5. Fig 1J: A line with R2 and p value for the correlation would be more informative.
6. Fig 2F: The regression is non-linear, but what equation is used?
7. Fig 3A: What is day 1 TRPV4 expression, does it increase in fast relaxing gels?
8. Fig 4C: Cartoon does not match with data in Fig.4D-G. For example, the Day 14 induction sample ('+D7 Induction') should probably be larger than Day 7 with the +OSM ('+D7 Ind. + Osm.') maintaining the same size given the data in Fig 4G.
9. Fig 4H: The inclusion of beta1 integrin is not well explained.

Reviewer #3:

Remarks to the Author:

In this manuscript, Lee et al utilize an alginate hydrogel to study the relationship between cell volume and osteogenic differentiation. The authors observe a strong correlation between cell volume and osteogenic differentiation when tuning cell volume by viscoelasticity, osmotic pressure or TRPV4 activity. The authors also elucidate the role of RUNX2, instead of YAP, involved in the molecular mechanism underlying this correlation. The topic and focus of this manuscript are interesting, however, the authors do not provide enough conclusive results to support their conclusions. This reviewer is especially concerned about the confounding effects within this complex system and the accuracy in the reported methodology, especially the measurement of cell volume and TRPV4 activity.

Major Concerns:

- 1) The measurement of cell volume is the most important and frequently used readout throughout the manuscript. Based on what the authors described in methods section, they take images of fixed and sectioned samples and use Imaris to calculate cell volume. Although this method has been previously used in literature, one recent paper (Xie, K., Yang, Y., & Jiang, H. (2018). Controlling Cellular Volume via Mechanical and Physical Properties of Substrate. Biophysical

journal, 114(3), 675-687.) found the inaccuracy of using this method to measure cell volume. Considering the importance of the accuracy of the cell volume measurement in this manuscript, this reviewer strongly suggests that the authors to confirm their volume measurement by other methods before drawing any conclusions.

2) In addition, another methodology that requires improvement is the quantification of YAP and RUNX2 nuclear localization. This reviewer suggests that the authors use a 3D reconstruction, similar to what they use for measuring the cell volume, to calculate the YAP and RUNX2 nuclear localization based on an intensity per volume. Also, the authors don't provide any details as to how they calculate the nuclear localization, based on a single slide or a maximum intensity projection?

3) The authors need to better characterize the TRPV4 activity by measuring the calcium levels or pre-labeling the calcium with a dye and monitoring the calcium dynamics in real time. Producing more TRPV4 protein doesn't necessary guarantee a higher activity of the ionic channel.

4) Although the authors explain the inconsistencies between their findings and published findings in 2D systems (Guo, M., Pegoraro, A. F., Mao, A., Zhou, E. H., Arany, P. R., Han, Y., ... & Mackintosh, F. C. Cell volume change through water efflux impacts cell stiffness and stem cell fate. *Proceedings of the National Academy of Sciences*, 201705179 (2017).) there are further inconsistencies with the relationship between cell volume and osteogenic differentiation, and the inconsistency in molecular mechanisms need to be addressed. In the outlook section of this manuscript, the authors themselves state that activation of the TRPV4 channel can activate calcium signaling induced RUNX2 nuclear translocation for osteogenic differentiation. However, this is counter to the previous literature that in 2D systems hMSCs show more osteogenic differentiation when the TRPV4 channel is inhibited. This reviewer is not convinced by the simple explanation that this behavior results from the difference between 2D and 3D culture, and requires more experiments and discussion to support the findings.

5) There are also many concerns about the complexity of the system and confounding effects that make it difficult to interpret some key results. First, it is confusing to this reviewer why the authors study the relationship between cell volume and osteogenic differentiation in a viscoelastic environment. Why is viscoelasticity needed? The authors don't provide any physiological or biological basis in the introduction. Similarly, the authors should provide more motivation as to why osmotic pressure should be used to control cell volume? Is osmotic pressure that different in the human body that cell volume would be tuned via this mechanism? Secondly, it is important to note that when tuning a cell's volume while maintaining its sphericity also affects the interactions between the cell and the surrounding matrix. A larger cell will also have a larger surface area and hence more interactions with the matrix. Similarly, a decrease in cell spreading can also reduce the interaction between cell and matrix; hence, the conclusions are less suspect if the authors cannot decouple the different parameters. As supported by numerous publications in the literature, the interaction and remodeling capacity of hMSCs in a 3D environment is as important, if not more important than, a cell volume change. In addition, local changes in the gel mechanics are known to be an important factor in viscoelastic materials, so how would the authors decouple the effect of mechanical changes versus volume changes.

6) The authors should also check cell stemness at the high osmotic pressure condition, as it might be possible that hMSCs are in an unhealthy state and this is why hMSCs express fewer osteogenic markers. In a healthy state, hMSC would express more stemness markers if less osteogenic markers are observed.

7) The only quantification of hMSC osteogenic differentiation in this manuscript is ALP staining, which significantly reduces the significance and credibility of the authors' findings. Additional characterization is required, for example, qPCR and western blots of different osteogenic markers (OCN, OPN, etc). Similarly, proliferation should be measured in differentiation conditions, as different cell densities are clearly observed in the representative images the author's selected, and this could likely affect osteogenic differentiation.

8) In figure 4h, the authors claim that $\beta 1$ integrin exhibited a similar distribution on the cell membrane with and without additional osmotic pressure. However, this statement needs to be supported by quantitative analysis, as it doesn't look similar to this reviewer. Similarly, I would recommend a correlation test among individual cells for Fig. 1d, as it could better support the authors' statements.

9) The authors should provide an explanation for the shape of the correlation curve between cell volume and ALP activity. Could this shape be predicted based on computational modeling? Why is it not linear or a sigmoidal growth curve? Without further support for the correlation curve, it would be better to tone down the language by simply saying cell volume and ALP activity are correlated with each other, as multiple types of a 'single curve' can be drawn based on the data.

Minor Concerns:

1) Many figures are cited incorrectly in the text, for example, a) on Page 3 Line 57, Supplementary Fig. 1a-c should be moved to Page 3 Line 59 aftera range of viscoelastic responses. b) Page 4 Line 86, it should be Fig. 1i instead of Fig. 1j. c) Page 6 Line 131, Supplementary Fig. 2b should be Supplementary Fig. 3b. and so on. All of the mislabeling was frustrating and left the impression of a poorly written manuscript.

2) Some words need to be changed and the authors need to tone down their language, as they overstate some of their conclusions, for example, Page 4 Line 78, to assess whether cell spreading was 'necessary' for cell volume expansion, Necessary need to be replaced, as not only cell spreading but other factors, such as osmotic pressure, can induce cell volume expansion. Page 5, line 109, what the authors reveal is only how the volume expansion occurs, and not the underlying molecular mechanism by which volume expansion promotes osteogenic differentiation.

RESPONSE TO REVIEWERS

We thank the reviewers for their thorough review of the manuscript, and their constructive criticisms and suggestions. We were pleased that all the reviewers were interested in our finding that cellular volume expansion and TRPV4 ion channel activation enhance osteogenesis of mesenchymal stem cells (MSCs) in 3D culture. Each reviewer has suggested important additional studies to support our findings and the proposed mechanism. To address all of these concerns and suggestions, we have conducted numerous new experiments and analyses, and have substantially revised the text (all major changes underlined). The revision experiments have reinforced our major findings, and significantly strengthened the manuscript. A specific point-by-point response follows, but we first highlight the major additions to the manuscript:

1. Given the challenge of precisely measuring cell volumes in 3D, the cell volume expansion results were confirmed with three sets of studies.
 - a. A comparison between the measured volume of fluorescent microbeads (diameter: 1 μm and 15 μm) encapsulated in hydrogels with different rates of stress relaxation was conducted. These measurements demonstrate that hydrogel formulation did not impact measurement of cell volume. These results are included in Supplementary Fig. S2a-c.
 - b. The effect of adjusting threshold values used for image processing was assessed. Varying the threshold values did not change the trend of cell volume expansion. These results are included in Supplementary Fig. S2d-f.
 - c. Finally, a super-resolution confocal microscopy technique (Zeiss Airyscan with deconvolution processing) was used as an alternative method to measure cell volumes. The super-resolution method improves axial resolution by 1.7x compared to standard confocal microscopy, with at least a 4-fold increase in signal-to-noise ratio (Korobchevskaya et al., *Photonics*, 2017). Comparison of both imaging techniques revealed that the measured volumes are consistent for both techniques. These results are included in Supplementary Fig. S2 g-h.
2. The impact of increasing cell volume, using hypoosmotic pressure, was assessed. These experiments revealed that cell volume expansion by hypoosmotic pressure in fast relaxing gels accelerates osteogenic differentiation. This provides independent support that cell volume expansion mediates osteogenic differentiation. These results are included in Fig. 2g-k.
3. Intracellular calcium levels and intracellular protein amounts were measured for cells cultured in hydrogels with different levels of stress relaxation or under varying osmotic pressures. The measurements indicated that cell volume expansion in fast relaxing hydrogels was associated with increased levels of calcium ions but not altered levels of intracellular proteins. These results are included in Fig. 3c, d, and Supplementary Fig. S8.
4. The effect of cell density on osteogenic differentiation was tested. Higher cell densities did not promote increased levels of osteogenic differentiation in slow relaxing gels or fast relaxing gels under high osmotic pressure. These indicate that increased proliferation was not the primary cause of increased osteogenesis in the fast relaxing hydrogels. These results are included in Supplementary Fig. S10.
5. Inhibition studies were conducted in order to elucidate the role of key molecules in mediating the impact of stress relaxation on differentiation. Pharmacological inhibition of myosin activity and actin polymerization restricted both cell volume expansion and osteogenic differentiation. Further, the ERK pathway was found to be involved in osteogenesis by mediating translocation of RUNX2 downstream of cell volume expansion in our system. These results are included in Fig. 5i-k and Supplementary Fig. S13.

Response to Reviewer #1

Lee et al reported an interesting work showing volume of MSC changes in 3-dimensional hydrogels upon different compositions and regulates stem cell fate. The finding is interesting and potentially important to

understand cell behavior in 3D. The paper is well written and understandable to a wide audience. Nonetheless, I'm not fully convinced based on the current submission due to the following reasons.

We appreciate that the reviewer found the study to be interesting and important for understanding cell behavior in 3D culture. We thank the reviewer for the constructive critiques.

My first major issue is about the volume measurement in 3D. The authors should report the accuracy of cell volume determinations. It will be possible to estimate the measurement uncertainty by calculating how the error of the fluorescence threshold value propagates to the cell volume. It is not clear how the threshold is obtained in the work, and what is the uncertainty of this threshold. Currently, the authors only state that the same threshold is used in this study. This does not tell us the uncertainty of this chosen threshold. How much does the calculated cell volume change when the threshold value is increased or decreased by the standard deviation?

We thank the reviewer for bringing up a critical point and an excellent suggestion. As the reviewer suggested, we have quantified cell volume with an adjusted threshold increased by the standard deviation. In addition, we used the optimized threshold values that the Imaris program automatically determines in each 3D-image to remove user bias. The trend of increased volume expansion in faster relaxing hydrogels is statistically significant using any of the thresholding methods. These results are now included in Supplementary Fig. S2d-f and are discussed in the manuscript. We note that decreasing the threshold by the standard deviation led to the entire imaging volume being included as signal. This is due to the asymmetric nature of the signal distribution.

The z resolution of the standard confocal microscope is approximately $1.4 \cdot \text{wavelength} \cdot \text{refractive index} / \text{NA}^2$ which is about 0.5 microns when respective parameters are 543nm, 1.4NA and in water. This axial resolution is the uncertainty of cell boundary localization in z-axis for an ideal case on a coverslip. In different gel compositions in 3D, the point spread function can significantly deviate from the one on the surface of a coverslip because gels can relax and swell, therefore the measurement uncertainty will be significantly affected and will also not be the same in different gel conditions. This can be clearly seen in figure 1a and 1e where the z-resolution seems ill in the spreading cell case. How much uncertainty in cell volume measurement due to this low and varying z-resolutions in this study should be examined and discussed.

The reviewer brings up several important points regarding measurement uncertainty in different gel conditions and z-resolution. To address these points, we first measured the volume of fluorescent microbeads encapsulated in different hydrogels to examine whether gel conditions affect the 3D volume measurements. The analysis demonstrated that all images of beads encapsulated in different hydrogels are similarly distorted in the z-direction and the error of the volume measurement by the limited z-resolution is in the range of 10% in all hydrogels. Importantly, the measured volumes of beads are not significantly different across all hydrogel conditions. Thus, the error in volume measurement results from fundamental limitations of optical microscopy, and not from differences in hydrogel conditions. Images of fluorescent microbeads and quantification of the bead volume encapsulated in different hydrogels are included in Supplementary Fig. S2a-c.

Next, we used an alternative method to measure 3D cell volume. Airyscan super-resolution confocal microscopy followed by deconvolution processing was used to improve z-resolution by 1.7X with an increase in signal-to-noise ratio of at least 4X (Korobchevskaya, *et al.*, *Photonics*, 2017). Cell volumes measured with the super-resolution technique are consistent with the results previously obtained using standard confocal microscopy. These results are included in Supplementary Fig. S2g-h and are discussed in the manuscript.

The cell volume expansion observed here presumably is interesting, however, no understanding is proposed. What content is changed, cellular water or proteins?

Further, how are the cell volume change and homeostasis achieved and maintained, at different extracellular matrix conditions? The authors seem to suggest that the viscoelastic relaxation allows the cell volume to expand. If so, this indicates that elastic matrix confinement is limiting cell volume in the slow relaxation gel. If this is the case, are there any elastic interaction between the matrix and the cell? In another word, is the elastic force from the matrix involved in setting the equilibrium cell volume? Or are the osmotic balance across cell membrane always maintained at different matrix relaxation conditions? If the osmotic balance is always maintained, as typically assumed, the 2 fold volume expansion induces the total amount of internal osmolytes to double. What are the dominating osmolytes, ions, small proteins, or ATP? Are the total number important for stem cell fate? These are critical for understanding cell volume expansion and their impact.

We thank the reviewer for the excellent suggestions. Regarding the source of cell volume expansion, we measured total intracellular protein and calcium concentration in cells encapsulated in hydrogels with different levels of stress relaxation and under different osmotic pressures. The protein concentration in single cells was not significantly different in all conditions (Supplementary Fig. S8), indicating that the major driver of cell volume changes is not alteration in protein amounts. However, calcium concentration in single cells was significantly different for cells in different relaxing hydrogels and under different osmotic pressure, which implicates changes in intracellular ion concentration as a major driver of cell volume difference. These results are included in Fig. 3c, d and are discussed in the manuscript.

The same questions continue in the osmotic compression condition. How stable is the cell volume under such non-isotonic condition? Again, if the osmotic balance is still maintained, the concentration of ions must increase in the cell to balance the additional osmotic pressure generated by PEG400 in the medium. If this is the case, the concentration of ions in the cell and that in the medium will not be balanced. In the meantime, various ions pumps are known to function if there is an imbalance in a particular type of ions. How is a dynamic osmotic balance still maintained in such cases?

The reviewer brings up a key point regarding stability of cellular volume under hyperosmotic pressure. We monitored the cell volume change over time when cells were cultured in fast relaxing hydrogels and under hyperosmotic pressure. We find that cell volumes increased over time under hyperosmotic pressure, but this increase of cell volume is significantly lower than the case of isotonic condition. Together with the data on calcium concentrations, these indicate that osmotic balance is maintained by an influx of calcium ions. These results are included in Supplementary Fig. S5.

It is claimed that volume expansion is driving the osteogenic differentiation, however, no controlled volume expansion experiments are performed. For example, the authors can increase cell volume by an alternative approach, perhaps hypotonic pressure, and show that osteogenic differentiation is still enhanced. The osmotic pressure experiment reducing cell volume is also not convincing enough as osmotic pressure can cause many issues to cell behavior and physiology, for example, can reduce cell proliferation significantly. Understanding the mechanism of cell volume expansion/change is therefore also important here since it will allow us to know what content is being regulated during volume expansion in the fast relaxation gel, therefore helps the design of control experiments to regulate cell volume to study its impact.

We thank the reviewer for an outstanding suggestion. As suggested, we induced cell volume expansion by applying hypoosmotic pressure and observed whether the volume expansion promotes rapid osteogenic differentiation at early time points. Cell volume expansion was accelerated in hypoosmotic conditions. Further, nuclear translocation of RUNX2 was significantly enhanced at early time points, which induces more rapid osteogenic differentiation, as indicated by significantly increased ALP activity at earlier timepoints. These results are now included in Fig. 2g-k, 5e and are discussed in the manuscript.

In addition, we have now assessed the role of proliferation in mediating osteogenic differentiation in the hydrogels. First, immunohistochemical stainings of Ki-67, a marker of proliferation, were performed. As expected, stress relaxation and osmotic pressure impacted proliferation of the MSCs. Slower relaxation and higher osmotic pressure both significantly diminish cell proliferation. Further, TRPV4 activation increases proliferation of MSCs and TRPV4 inhibition decreases proliferation of MSCs. To test the role of altered cell density, arising from different levels of proliferation, in mediating osteogenic differentiation, MSCs were seeded at higher densities in slow relaxing hydrogels or in fast relaxing hydrogels under increased osmotic pressure. Importantly, higher cell seeding density in slow relaxing hydrogels or in fast relaxing hydrogels under increased osmotic pressure did not increase osteogenic differentiation and even significantly decreased osteogenesis. These demonstrate that enhanced osteogenesis in fast relaxing hydrogels is not mediated by increased cell proliferation. These results are now included in Supplementary Fig. 10. and are noted in the manuscript.

Also, cell morphology changes significantly upon cell spreading, suggests that nuclear morphology could change therefore nuclear lamin expression could also change. Similarly, under osmotic compression, cell nuclear size and morphology must change, therefore, can also affect nuclear lamin expression. As nuclear lamin has been demonstrated repeatedly to play a role in stem cell fate (reference 6 in the manuscript), the status and role of nuclear lamin should be commented.

We thank the reviewer for the suggestion and agree that nuclear lamins could play a significant role in stem cell differentiation in our system. Western blots were performed to assess lamin A/C protein levels, previously shown in the cited reference to change in response to matrix stiffness. However, no significant changes in lamin A/C protein levels with respect to osmotic pressure or stress relaxation were observed. The data are included in Supplementary Fig. S12.

Another issue is that cell volume change is associated with cell spreading. Even under additional osmotic pressure, as the authors claim that cell spreading is not affected, what about cell surface area or other morphological factors? My question is that how could the authors be sure that the observed effect in stem cell fate is due to cell volume, not cell surface area or the number of branches? I find it unclear to conclude that cell volume expansion is the cause for stem cell fate change. What the authors found is potentially important, however, it is unclear if cell volume is the cause here, not merely a consequence.

We thank the reviewer for raising this interesting point. Our additional osmotic pressure experiments, shown in Fig. 4, demonstrate that cell spreading and number of branches does not direct osteogenic differentiation. Increased osmotic pressure did not impact sphericity, related to cell spreading and number of branches, but did lead to a decrease in cell volume and osteogenesis. However, changes in cell volume in this experiment should be associated with changes in cell surface area. To assess the potential role of cell surface area, we compared cell surface area, volume, and osteogenesis for MSCs cultured first in growth media and then in induction media with increased osmotic pressure with MSCs cultured in induction media (Supplementary Fig. S11). While the surface area of cells cultured in the growth media then induction media/osmotic pressure were similar to that of cells cultured in induction media, cell volume and levels of osteogenesis were substantially higher when MSCs were cultured directly in induction media. These indicate that surface area does not control stem cell fate, and implicate cell volume as the key controlling factor.

Minor:

1. During encapsulation, what is the osmotic pressure of the solution including medium and polymers and ions? Is there any volume increase or decrease during encapsulation?

We compared cell volume before encapsulation and after encapsulation. There is no significant cell volume change. This result is now included in Supplementary Fig. S15.

2. Clear YAP localization change under varying osmotic pressures has recently been observed by several different groups, for example in several works by Kunliang Guan's group, and another one by Alberto Elosegui-Artola et al, Cell, 2017. YAP/TAZ are known to regulate stem cell fate. How do we understand the difference in the YAP localizations in those studies and in the current one?

We agree that YAP nuclear localization under osmotic pressure has been observed by several groups. However, all of these observations have been based on 2D cell culture environment, not 3D microenvironment in which our results are observed. A previous study (Caliari et al., *Biomaterials* 2016) showed that YAP nuclear localization is strongly dependent on dimensionality of cell culture environment.

3. What is the role of matrix stiffness here? As reference 3 in the current manuscript showed, decreasing matrix stiffness causes less osteogenesis.

As in reference 3, we confirmed that osteogenic differentiation indicated by ALP level in soft hydrogels was significantly less than that in stiff hydrogels. The result is now included in Supplementary Fig. S14.

4. What exactly is the role of TRPV4 and how they are involved in volume regulation is not clear from the current text.

Many previous studies have determined the relationship between cell volume and TRPV4, showing that TRPV4 plays a major role in cell volume regulation by allowing calcium ions into the cell to balance intracellular and extracellular osmotic pressure. We apologize for the unclear wording of the text. We have now revised the text to better explain the role of TRPV4 in volume regulation.

Original description

“Previous studies have demonstrated that the transient receptor potential vanilloid-4 (TRPV4) ion channels were directly activated by membrane stretching¹⁷⁻²⁰ or forces applied by β 1 integrin membrane receptor²¹, and that cell volume could be regulated by TRPV4 activity^{22,23}, making it a logical candidate for study.”

Revised description

“Previous studies have demonstrated that the transient receptor potential vanilloid-4 (TRPV4) ion channels were directly activated by membrane stretching²²⁻²⁵ or forces applied by β 1 integrin membrane receptor²⁶, and TRPV4 regulates cell volume through balancing osmolality of calcium ions in the cytoplasm^{27,28}, making it a logical candidate for study.”

Response to Reviewer #2

The authors describe an apparent correlation between cell volume and efficiency of MSC osteogenic differentiation in 3D alginate culture that seems to depend on TRPV4 activation. First, MSCs were cultured in 20kPa alginate gels with varying relaxation times, where gels with fast relaxation time allows for cell spreading (increase of cell volume) and effective osteogenic differentiation. Inhibition of cell spreading by hyperosmotic challenge (by addition of PEG) seems to reduce the osteogenic differentiation efficiency of cells seeded in gels with fast relaxation time. The effects can be recapitulated by TRPV4 inhibition and rescued by TRPV4 activation. Additionally, by immunostaining for RUNX2, hyperosmotic stress and TRPV4 regulation also affect RUNX2 translocation, which is essential for osteogenic differentiation. Considering the growing reports on the role of TRPV4 in mechanotransduction and the lack of knowledge in how MSCs perceive 3D environments, this study is timely and will expand the field. The data of this study is clear and the role of TRPV4 is apparent. However, a few concerns temper enthusiasm:

We appreciate that the reviewer found this work to be timely and capable of expanding the field of mechanotransduction, and thank the reviewer for the constructive critiques.

1. New bone is typically formed in vivo by MSC-generated osteoblasts on top of osteoclast-resorbed bone, which is a 2-dimensional process. In what precise situations is this 3D approach relevant to the biology of bone? Further, is there reason to believe that in bone, osmolality differs in ways studied here? And when will cell volume regulation play a role in such process? How do the mechanical parameters of these gels compare to that of osteoid?

The reviewer brings up an excellent point, and we agree that providing an appropriate biological context for our study is important. MSCs do differentiate into osteoblasts on top of osteoclast-resorbed bone *in vivo*, indicating the physiological relevance of 2D stem cell differentiation studies. However, we expect that a 3D viscoelastic microenvironment is relevant to *in vivo* osteogenic differentiation of MSCs during bone fracture healing. In a previous study, Chaudhuri and colleagues found that a fracture hematoma, which contain MSCs in a three-dimensional context during healing of bone fractures, exhibits fast stress relaxation (Chaudhuri et al., *Nature Materials* 2016). Further, the results of our studies are immediately relevant to biomaterials-based approaches in regenerative medicine, in which MSCs are encapsulated into 3D hydrogel scaffolds. The physiological relevance of 2D and 3D culture to stem cell differentiation is now discussed in the introduction section of the manuscript.

Regarding the point about osmolality, varying osmotic pressure was used as a method to independently modulate cell volume, and not to simulate conditions that might occur physiologically.

2. How do the mechanical parameters of these gels vary in space and time around the cells?

The reviewer brings up an interesting question. Previous studies have reported that alginate hydrogels used in this study are stable and non-degradable over a week (Chaudhuri et al., *Nature Materials* 2016 and Lee et al., *Nature Materials* 2017). However, cells could mechanically remodel their local mechanical microenvironment, changing the local stiffness and stress relaxation. Unfortunately, there are no established techniques for measuring mechanical properties of microenvironments in 3D matrix in space and time at the microscale in live cell-culture experiments. This represents a substantial challenge in the field of materials science. It would be interesting and important to develop such a technique, but this would be outside the scope of the present manuscript. We have acknowledged this point in the discussion section of the manuscript.

3. Cell volume has long been known to be regulated by cytoskeleton (eg. cytochalasin) and vice versa. Decoupling should be examined. TRPV4's role in regulating cell contractility has also been widely reported,

given its primary function as a calcium channel. What is the effect of TRPV4 regulation and prolonged hyperosmotic challenge on actomyosin organization and contractility.

We thank the reviewer for bringing up an important point and suggestion. We have now examined cell volume expansion in the fast relaxing hydrogel in the presence of pharmacological inhibitors of myosin light chain kinase or MLCK (with ML-7), actin polymerization (with Cytochalasin D), or myosin II (with blebbistatin). We find that inhibition of MLCK, actin polymerization, and myosin II significantly reduces both volume expansion and osteogenesis in hydrogels with fast stress relaxation. These indicate actomyosin contractility and actin polymerization to be necessary for cell volume expansion and osteogenesis. These results are now included in Supplementary Fig. S13 and are discussed in the manuscript.

4. Seeding in gels with slow relaxation time, hyperosmotic media and TRPV4 inhibition all seem to decrease cell count (per images), while TRPV4 activation does the opposite. Given the likely importance of cell density in osteogenic differentiation, several things need to be checked for all the conditions:

a. Effects of seeding density (e.g. does seeding more cells in slow relaxation time gel alter ALP production?) or are all of these single cells truly isolated?

b. Effects of TRPV4 regulation and hyperosmotic media on cell proliferation

We thank the reviewer for the suggestion of examining the role of seeding density and proliferation in mediating the impact of stress relaxation and osmotic pressure on differentiation. First, immunohistochemical stainings of Ki-67, a marker of proliferation, were performed. As expected, stress relaxation and osmotic pressure impacted proliferation of the MSCs. Slower relaxation and higher osmotic pressure both significantly inhibit cell proliferation. Further, TRPV4 activation increases proliferation of MSCs and TRPV4 inhibition decreases proliferation of MSCs. To test the role of altered cell density, arising from different levels of proliferation, in mediating osteogenic differentiation, MSCs were seeded at higher densities in slow relaxing hydrogels or in fast relaxing hydrogels under increased osmotic pressure. Importantly, higher cell seeding density in slow relaxing hydrogels or in fast relaxing hydrogels under increased osmotic pressure did not increase osteogenic differentiation and even significantly decreased osteogenesis. These demonstrate that enhanced osteogenesis in fast relaxing hydrogels is not mediated by increased cell proliferation. These results are now included in Supplementary Fig. 10. and are noted in the manuscript.

5. It is not clear from the citation provided (33) what the role of nuclear translocation of RUNX2 is. This should be more fully explored here, including RUNX2's phosphorylation.

We thank the reviewer for pointing out this discrepancy and the suggested experiments. We have now cited a more relevant reference that demonstrated that RUNX2 nuclear localization is necessary for osteogenesis (Kim et al., *Genes and Dev.*, 2003). RUNX2 has many potential phosphorylation sites and the precise site or sites that are phosphorylated to induce nuclear translocation are not known. Further, commercial phospho-specific antibodies are not available for the majority of reported sites. However, the ERK pathway is known to phosphorylate RUNX2. Pharmacological inhibition of ERK1/2 led to a significant decrease in ALP-positive cells and nuclear RUNX2, implicating ERK1/2 in RUNX2 activation. Interestingly, we did not observe a change in cell volume upon ERK inhibition, indicating that ERK activation of RUNX2 is downstream of TRPV4-induced volume expansion. These results are included in Fig. 5 and discussed in the manuscript.

6. Fig.3: What are the contributions of TRPV4 activity vs TRPV4 expression levels (abundance, Fig.3A) vs TRPV4 localization /clustering (Fig.3B) in volume regulation, differentiation, etc.? What aspect of TRPV4 is most important here?

The reviewer brings up an interesting question regarding the function of TRPV4, as both TRPV4 localization and expression levels could influence activity. However, it is difficult to decouple those contributions, as they are both changing in our system. Nonetheless, we conducted calcium imaging to measure the level of intracellular calcium at day 7 to demonstrate differences in TRPV4 function. Calcium concentration is significantly higher for MSCs in faster relaxing hydrogels and under less osmotic pressure, indicating TRPV4 expression level and clustering are highly correlated to TRPV4 function. These results are now included in Fig. 3 and are discussed in the manuscript.

Minor points:

1. *Fig. 1E is missing arrows that are mentioned in the figure caption.*

Arrows have been added.

2. *Fig. 5: labels for the images are missing (e.g. blue is Nucleus)*

Color labels have been added in the images.

3. *Please include the conversion of kPa osmotic pressure to mOsm/kg or mOsm/L (unit of osmolality) in the method section.*

Osmotic pressures are now also reported in units of osmolality: 6%wt/vol (150 mOsm/L), 3% (75 mOsm/L), and 1.5% (37.5 mOsm/L)

4. *Fig 1D: Plot is probably not needed with the data present in b & c.*

While Figure 1b and c are sufficient to indicate a correlation between cell volume and sphericity, the revised figure 1D with a statistical test more clearly and directly displays the correlation.

5. *Fig 1J: A line with R2 and p value for the correlation would be more informative.*

R² and p values have been added for all correlation data.

6. *Fig 2F: The regression is non-linear, but what equation is used?*

We used a method of the least square fit to find the best fit-curve to capture the strong correlation between cell volume and ALP activity. The equation of the curve is $Y = (-334.7)e^{-0.00115X} + 108.7$.

7. *Fig 3A: What is day 1 TRPV4 expression, does it increase in fast relaxing gels?*

We found that the Day 1 TRPV4 expression was not significantly different from cells in Day 7 in fast relaxing gels. We have updated the description of TRPV4 expression in the manuscript to describe the expression as decreasing in slow relaxing hydrogels or in cells under hyperosmotic pressure. These results are presented in Supplementary Fig. S7.

8. *Fig 4C: Cartoon does not match with data in Fig.4D-G. For example, the Day 14 induction sample ('+D7 Induction') should probably be larger than Day 7 with the +OSM ('+D7 Ind. + Osm.')*
maintaining the same size given the data in Fig 4G.

The cartoon has been revised to match the data.

9. *Fig 4H: The inclusion of beta1 integrin is not well explained.*

As this information may be confusing, we have removed the figures and revised the text.

Response to Reviewer #3

In this manuscript, Lee et al utilize an alginate hydrogel to study the relationship between cell volume and osteogenic differentiation. The authors observe a strong correlation between cell volume and osteogenic differentiation when tuning cell volume by viscoelasticity, osmotic pressure or TRPV4 activity. The authors also elucidate the role of RUNX2, instead of YAP, involved in the molecular mechanism underlying this correlation. The topic and focus of this manuscript are interesting, however, the authors do not provide enough conclusive results to support their conclusions. This reviewer is especially concerned about the confounding effects within this complex system and the accuracy in the reported methodology, especially the measurement of cell volume and TRPV4 activity.

We appreciate that the reviewer found the topic and focus of this study to be interesting. We thank the reviewer for the constructive critiques.

Major Concerns:

1) The measurement of cell volume is the most important and frequently used readout throughout the manuscript. Based on what the authors described in methods section, they take images of fixed and sectioned samples and use Imaris to calculate cell volume. Although this method has been previously used in literature, one recent paper (Xie, K., Yang, Y., & Jiang, H. (2018). Controlling Cellular Volume via Mechanical and Physical Properties of Substrate. Biophysical journal, 114(3), 675-687.) found the inaccuracy of using this method to measure cell volume. Considering the importance of the accuracy of the cell volume measurement in this manuscript, this reviewer strongly suggests that the authors to confirm their volume measurement by other methods before drawing any conclusions.

The reviewer brings up an important point. We have now conducted three different sets of studies to further support our volume measurement conclusions, and discussed these results in the manuscript. Firstly, we have quantified cell volume with an adjusted threshold increased by the standard deviation (as suggested by Reviewer 1), or optimized threshold values that the Imaris program automatically determines for each 3D-image. Based on these results, we have found that the trend of volume expansion in faster relaxing hydrogels is statistically significant even with different threshold values. These results are included in Supplementary Fig. S2a-c.

Secondly, we also measured the volume of fluorescent microbeads encapsulated in hydrogels with different rates of stress relaxation to examine whether gel conditions systematically affected the volume measurements. The analysis demonstrated that all images of beads encapsulated in different hydrogels are equally distorted along the z-direction due to technical limitations of confocal microscopy and that the error of the volume measurement is ~10% for all hydrogels. Importantly, the measured bead volumes were not significantly different between any of the hydrogel conditions. These results are included in Supplementary Fig. S2d-f.

Lastly, we used an alternative method to measure 3D cell volume. Airyscan super-resolution confocal microscopy followed by deconvolution processing was used to improve z-resolution by 1.7X with an increase in signal-to-noise ratio of at least 4X (Korobchevskaya, *et al.*, *Photonics*, 2017). Cell volumes measured with the super resolution technique are consistent with volumes measured using standard confocal microscopy. These results are included in Supplementary Fig. S2g-h.

2) In addition, another methodology that requires improvement is the quantification of YAP and RUNX2 nuclear localization. This reviewer suggests that the authors use a 3D reconstruction, similar to what they use for measuring the cell volume, to calculate the YAP and RUNX2 nuclear localization based on an

intensity per volume. Also, the authors don't provide any details as to how they calculate the nuclear localization, based on a single slide or a maximum intensity projection?

We have now included the comparison between 2D and 3D quantification of YAP and RUNX2 nuclear localization in supplementary Fig. S16. These results demonstrate that 2D quantification of nuclear localization based on random sectioned images are similar to 3D quantification of nuclear localization. Further, we have added methodological detail describing how nuclear localization and the nuclear-to-cytoplasmic ratio were determined.

3) The authors need to better characterize the TRPV4 activity by measuring the calcium levels or pre-labeling the calcium with a dye and monitoring the calcium dynamics in real time. Producing more TRPV4 protein doesn't necessary guarantee a higher activity of the ionic channel.

We thank the reviewer for the outstanding suggestion. Calcium imaging was conducted to measure the level of intracellular calcium at day 7. The experiments revealed that the level of calcium in single cells is significantly higher in faster relaxing hydrogels, indicating higher TRPV4 activity in the fast relaxing hydrogels. Calcium images and the quantification are included in Fig 3, and the results are discussed in the manuscript.

4) Although the authors explain the inconsistencies between their findings and published findings in 2D systems (Guo, M., Pegoraro, A. F., Mao, A., Zhou, E. H., Arany, P. R., Han, Y., ... & Mackintosh, F. C. Cell volume change through water efflux impacts cell stiffness and stem cell fate. Proceedings of the National Academy of Sciences, 201705179 (2017).) there are further inconsistencies with the relationship between cell volume and osteogenic differentiation, and the inconsistency in molecular mechanisms need to be addressed. In the outlook section of this manuscript, the authors themselves state that activation of the TRPV4 channel can activate calcium signaling induced RUNX2 nuclear translocation for osteogenic differentiation. However, this is counter to the previous literature that in 2D systems hMSCs show more osteogenic differentiation when the TRPV4 channel is inhibited. This reviewer is not convinced by the simple explanation that this behavior results from the difference between 2D and 3D culture, and requires more experiments and discussion to support the findings.

We thank the reviewer for asking about the consistency of our observation with previous studies. While the reviewer mentioned previous literature using 2D systems showing hMSCs undergoing more osteogenic differentiation when TRPV4 channels are inhibited, we found several papers that observed the opposite result, or that TRPV4 activation induces osteogenic differentiation of MSCs, consistent with our findings. These include the following:

1. O'Connor, et al. "Increased susceptibility of Trpv4-deficient mice to obesity and obesity-induced osteoarthritis with very high-fat diet." *Annals of the rheumatic diseases* 72.2 (2013): 300-304. This paper reported that MSCs from TRPV4 knockout mice had decreased osteogenic differentiation potential.
2. Hu, et al. "TRPV4 functions in flow shear stress induced early osteogenic differentiation of human bone marrow mesenchymal stem cells." *Biomedicine & Pharmacotherapy* 91 (2017): 841-848. This paper showed that TRPV4 activation by fluid shear stress induced early osteogenic differentiation of hMSCs.
3. Corrigan, et al. "TRPV4-mediates oscillatory fluid shear mechanotransduction in mesenchymal stem cells in part via the primary cilium." *Scientific reports* 8.1 (2018): 3824. This paper demonstrated that TRPV4 localization to the primary cilium by fluid shear stress enhanced early osteogenic differentiation of MSCs.

These previous studies are consistent with our study and also observe this phenomenon in a 3D environment.

5a) There are also many concerns about the complexity of the system and confounding effects that make it difficult to interpret some key results. First, it is confusing to this reviewer why the authors study the relationship between cell volume and osteogenic differentiation in a viscoelastic environment. Why is viscoelasticity needed? The authors don't provide any physiological or biological basis in the introduction. Similarly, the authors should provide more motivation as to why osmotic pressure should be used to control cell volume? Is osmotic pressure that different in the human body that cell volume would be tuned via this mechanism?

We thank the reviewer for the insightful comments and agree that physiological context is important. In a previous study, Chaudhuri and colleagues found that fracture hematoma, which contain MSCs in a three-dimensional context during healing of bone fractures, exhibits fast stress relaxation (Chaudhuri et al., *Nature Materials* 2016). We have emphasized this physiological context in the manuscript to clarify this point. Since the focus of this study is to elucidate the impact of stress relaxation on cell volume and how cell volume affect differentiation of MSCs in 3D culture, we applied varying osmotic pressure only as a method to independently modulate cell volume. We also highlight that these findings will be valuable to the tissue engineering and regenerative medicine communities as well, since directing differentiation in 3D environments is a key goal.

Regarding the point about osmotic pressure, varying osmotic pressure was used as a method to independently modulate cell volume, and not to simulate conditions that might occur physiologically.

5b) Secondly, it is important to note that when tuning a cell's volume while maintaining its sphericity also affects the interactions between the cell and the surrounding matrix. A larger cell will also have a larger surface area and hence more interactions with the matrix. Similarly, a decrease in cell spreading can also reduce the interaction between cell and matrix; hence, the conclusions are less suspect if the authors cannot decouple the different parameters. As supported by numerous publications in the literature, the interaction and remodeling capacity of hMSCs in a 3D environment is as important, if not more important than, a cell volume change. In addition, local changes in the gel mechanics are known to be an important factor in viscoelastic materials, so how would the authors decouple the effect of mechanical changes versus volume changes.

We agree that other morphological factors could be important for stem cell fate. Our additional osmotic pressure experiments, shown in Fig. 4, demonstrate that cell spreading does not direct osteogenic differentiation. Increased osmotic pressure did not impact sphericity, related to cell, but did lead to a decrease in cell volume and osteogenesis. However, changes in cell volume in this experiment should be associated with changes in cell surface area. To assess the potential role of cell surface area, we compared cell surface area, volume, and osteogenesis for MSCs cultured first in growth media and then in induction media with increased osmotic pressure with MSCs cultured in induction media (Supplementary Fig. S11). While the surface area of cells cultured in the growth media then induction media/osmotic pressure were similar with that of cells cultured in induction media, cell volume and levels of osteogenesis were substantially higher when MSCs were cultured directly in induction media. These indicate that surface area does not control stem cell fate, and implicate cell volume as the key controlling factor.

Further, we agree that there could be local changes in gel mechanics due to cell remodeling activity, and discuss this point in the discussion section of the manuscript. However, the osmotic pressure experiments, particularly those when osmotic pressure is changed after 2 days of culture (Fig 2g-k) or 7 days of culture (Fig. 4c-h), indicate cell volume as a key controlling factor, since changes in osmotic pressure directly impact cell volume but do not directly impact matrix mechanics.

6) The authors should also check cell stemness at the high osmotic pressure condition, as it might be

possible that hMSCs are in an unhealthy state and this is why hMSCs express fewer osteogenic markers. In a healthy state, hMSC would express more stemness markers if less osteogenic markers are observed.

We thank the reviewer for this suggestion. We performed immunofluorescent staining for CD105, a key stemness marker, and observed robust staining for CD105 in cells under high osmotic pressure and in hydrogels with slow stress relaxation. As the reviewer commented, this is the expected trend for conditions that induce less osteogenesis, and confirms that the MSCs are not in an unhealthy state. These results are now included in Supplementary Fig. S6.

7) The only quantification of hMSC osteogenic differentiation in this manuscript is ALP staining, which significantly reduces the significance and credibility of the authors' findings. Additional characterization is required, for example, qPCR and western blots of different osteogenic markers (OCN, OPN, etc). Similarly, proliferation should be measured in differentiation conditions, as different cell densities are clearly observed in the representative images the author's selected, and this could likely affect osteogenic differentiation.

To further support the osteogenic differentiation results, we assessed osteocalcin expression by Western blot and found that this marker follows the same trend that we observed with ALP staining and quantification. This data is included in Supplementary Fig. S4.

In addition, we have now assessed the role of proliferation in osteogenic differentiation in the hydrogels. First, immunohistochemical stainings of Ki-67, a marker of proliferation, were performed. As expected, stress relaxation and osmotic pressure impacted proliferation of the MSCs. Slower relaxation and higher osmotic pressure both significantly inhibit cell proliferation. Further, TRPV4 activation increases proliferation of MSCs and TRPV4 inhibition decreases proliferation of MSCs. To test the role of altered cell density, arising from different levels of proliferation, in mediating osteogenic differentiation, MSCs were seeded at higher densities in slow relaxing hydrogels or in fast relaxing hydrogels under increased osmotic pressure. Importantly, higher cell seeding density in slow relaxing hydrogels or in fast relaxing hydrogels under increased osmotic pressure did not increase osteogenic differentiation and even significantly decreased osteogenesis. These demonstrate that enhanced osteogenesis in fast relaxing hydrogels is not mediated by increased cell proliferation. These results are now included in Supplementary Fig. 10. and are noted in the manuscript.

8) In figure 4h, the authors claim that $\beta 1$ integrin exhibited a similar distribution on the cell membrane with and without additional osmotic pressure. However, this statement needs to be supported by quantitative analysis, as it doesn't look similar to this reviewer. Similarly, I would recommend a correlation test among individual cells for Fig. 1d, as it could better support the authors' statements.

The reviewer brings up a valid critique regarding the analysis of images and a correlation test for Fig. 1d. As this information may confuse readers, we have removed the figure and revised the text accordingly.

9) The authors should provide an explanation for the shape of the correlation curve between cell volume and ALP activity. Could this shape be predicted based on computational modeling? Why is it not linear or a sigmoidal growth curve? Without further support for the correlation curve, it would be better to tone down the language by simply saying cell volume and ALP activity are correlated with each other, as multiple types of a 'single curve' can be drawn based on the data.

The correlation curve was empirically obtained through a least square fit method with the paired data between representative cell volume and ALP activity obtained under different relaxation properties, RGD concentrations, and osmotic pressures. We found that all the other results from TRPV4 inhibition/activation experiments follow the single curve very well. As suggested, we have toned down the language by simply

saying cell volume and ALP activity are highly correlated with each other.

Minor Concerns:

1) Many figures are cited incorrectly in the text, for example, a) on Page 3 Line 57, Supplementary Fig. 1a-c should be moved to Page 3 Line 59 aftera range of viscoelastic responses. b)Page 4 Line 86, it should be Fig. 1i instead of Fig. 1j. c)Page 6 Line 131, Supplementary Fig. 2b should be Supplementary Fig. 3b. and so on. All of the mislabeling was frustrating and left the impression of a poorly written manuscript.

We sincerely apologize for incorrect citations of figures in manuscript. All the figure call-outs are now correctly cited.

2) Some words need to be changed and the authors need to tone down their language, as they overstate some of their conclusions, for example, Page 4 Line 78, to assess whether cell spreading was 'necessary' for cell volume expansion, Necessary need to be replaced, as not only cell spreading but other factors, such as osmotic pressure, can induce cell volume expansion. Page 5, line 109, what the authors reveal is only how the volume expansion occurs, and not the underlying molecular mechanism by which volume expansion promotes osteogenic differentiation.

The expressions have been revised as suggested.

References:

- [1] Korobchevskaya, Kseniya, et al. "Exploring the potential of airyscan microscopy for live cell imaging." *Photonics*. Vol. 4. No. 3. Multidisciplinary Digital Publishing Institute, 2017.
- [2] Caliari, Steven R., et al. "Dimensionality and spreading influence MSC YAP/TAZ signaling in hydrogel environments." *Biomaterials* 103 (2016): 314-323.
- [3] Chaudhuri, Ovijit, et al. "Hydrogels with tunable stress relaxation regulate stem cell fate and activity." *Nature materials* 15.3 (2016): 326.
- [4] Lee, Hong-pyo, et al. "Mechanical confinement regulates cartilage matrix formation by chondrocytes." *Nature materials* 16.12 (2017): 1243.
- [5] Kim, Sunhwa, et al. "Stat1 functions as a cytoplasmic attenuator of Runx2 in the transcriptional program of osteoblast differentiation." *Genes & development* 17.16 (2003): 1979-1991.
- [6] O'conor, Christopher J., et al. "Increased susceptibility of Trpv4-deficient mice to obesity and obesity-induced osteoarthritis with very high-fat diet." *Annals of the rheumatic diseases* 72.2 (2013): 300-304.
- [7] Hu, Kongzu, et al. "TRPV4 functions in flow shear stress induced early osteogenic differentiation of human bone marrow mesenchymal stem cells." *Biomedicine & Pharmacotherapy* 91 (2017): 841-848.
- [8] Corrigan, Michele A., et al. "TRPV4-mediates oscillatory fluid shear mechanotransduction in mesenchymal stem cells in part via the primary cilium." *Scientific reports* 8.1 (2018): 3824.

Reviewers' Comments:

Reviewer #1:

Remarks to the Author:

Lee et al. have made a significant amount of effort addressing mine and other referees' concerns. I am fine with the volume measurements itself now with the added comparison with Zeiss super-resolution and detailed measurements of optical bias in different matrices. The added calcium imaging and TRPV4 clarification are also interesting and consistent with several previous literature showing that TRPV4 and calcium signaling are responsible for osteogenesis, as the authors pointed out. However, I still don't think that the current experimental results presented in the revised manuscript are enough to claim that cell volume expansion is a critical factor here. Instead, it is very possible that TRPV4 activation is the key here, which induces both osteogenesis and volume change. I still think the experimental data itself is very interesting, however, I am not convinced by the current argument.

1. As the authors described in their response to my previous comment "calcium concentration in single cells was significantly different for cells in different relaxing hydrogels and under different osmotic pressure, which implicates changes in intracellular ion concentration as a major driver of cell volume difference", the observed volume change might be a consequence of the TRPV4 activation and ion concentration change, instead of a driving force of osteogenesis. The data certainly show that fast relaxing gel allows cell volume expansion, and results in TRPV4 activation. However, TRPV4 activation itself may induce both osteogenesis, as the authors pointed out, and cell volume expansion, also as the authors argue. In fact the change in TRPV4 protein expression level and calcium amount are consistent with the level of osteogenesis under various conditions. So the question becomes "is volume expansion necessary?" I don't see any current data isolating these effects and to show that volume change is required in regulating osteogenesis.

2. A possible role of volume expansion however could be mechanically triggering further relaxation of the matrix. As hypertonic pressure is applied, cells shrink to allow the matrix to relax and reorganize around the cell and confine them. Then it becomes similar to a slow relaxing matrix, from the perspective of cell confinement. If a hypotonic pressure is applied, cells expand to force the surrounding matrix to relax. Then this pressure essentially facilitates the gel to relax and allow cell spreading. I think matrix relaxation might actually be the key factor here.

3. The authors claim that calcium ions are responsible for changing the osmotic pressure balance and thus regulate cell volume. However, the calcium ion imaging is not quantitative. Calcium ions are indeed critical in cell physiology as they trigger many key functions such as muscle contraction, signaling, heart beating. However, it is not as concentrated as other ions such as Na and K in the cellular environment. How much osmotic pressure difference can be generated by Calcium channels moving Calcium ions along? The authors observed a volume change of 2 folds. This suggests that the amount of ions changed 2 fold in different matrices. It is unlikely to be dominated by Calcium ions given a low concentration is typically required for its functionality.

Reviewer #2:

Remarks to the Author:

The revision is only partially responsive to previous comments, and the authors should also be more precise in their explanations (a couple of examples are given below but elsewhere also).

1. The authors need to write in the main text their response: "varying osmotic pressure was used as a method to independently modulate cell volume, and not to simulate conditions that might occur physiologically." Secondly, although they added in Introduction some words about in vivo osteogenesis being 2D, they must change the word "necessary" in describing 3D because even Ref.11 shows some osteogenesis in the absence of matrix remodeling: "matrix remodeling is necessary for osteogenic differentiation, and can occur through either protease-mediated degradation or physical remodeling of matrices that are viscoelastic and exhibit fast stress relaxation 11"

2. The authors provide no data to address the question: How do the mechanical parameters of these gels vary in space and time around the cells? In their previous study (Chaudhuri et al., Nature Materials 2016), fast relaxing polymer accumulated around cells, which would imply a local stiffening. Does that occur in the present studies and when in relation to differentiation?

3. Questions about contractility and TRPV4 were poorly addressed. Contractility has long been shown to be pro-osteogenic in 2D (eg. Ref.1), and the authors now confirm this for their 3D (as should be stated in main text) together with cell volume measurements in new Fig.S13, but they did not address: "What is the effect of TRPV4 regulation ... on actomyosin organization and contractility." Given the new data showing (as expected) that contractility inhibitors suppress cell volume and osteogenesis, it is now critical for the authors to do another experiment that addresses whether their TRPV4 agonist (which enhanced osteogenic differentiation in hydrogels ... with an intermediate stress relaxation) over-rides the inhibitory effects of blebbistatin. This will address whether TRPV4 is simply upstream of contractility, which is a pathway that is established in the literature (per previous review comment). The authors must also explain in the main text the relevance of their results to osteogenesis in TRPV4 knockout mice such as those studied years ago in: "Impaired pressure sensation in mice lacking TRPV4. Suzuki M et al. J Biol Chem. 2003"

4. The added proliferation data for TRPV4 and the other conditions suggests proliferation is key to enhanced differentiation. This needs to be examined and discussed.

Reviewer #3:

Remarks to the Author:

In this revision, the authors have been highly responsive to the original critiques, addressing many of the technical critiques of the experimental methods and analysis of the results. At this stage, some questions remain (detailed below), but I am less certain about the physiological relevance of the studies. The authors have attempted to address this in their revision, but it is not convincing. The results may likely be observations that occur when MSCs are cultured in a narrow subset of hydrogels used by bioengineers (alginates that also bind calcium) and that the findings may be less applicable to any observations of MSCs in vivo. While the reviewer values that MSCs are used for many regenerative medicine applications and these results could assist in the design of biomaterials scaffolds for them, it is less clear that this will be of broader interest and it may be better suited for a more specialized tissue engineering or biomaterials journal audience. However, I trust the editor to make this final decision.

1. The MSCs were cultured in osteogenic and adipogenic induction media (mixed media), so my comment remains that the MSC characterization is weak. Certainly key markers of osteogenic differentiation are reported, but this should be strengthened and multiple markers at the gene and protein level reported. Also, given the media used, adipogenic markers should be reported.

2. I have been unable to find any reports in the literature where such a dramatic volume change occurs in MSCs in vivo. What is the physiological relevance of the volume change? Answering this question would help improve the impact of this work. For example, during cardiac hypertrophy and disease progression, the size of the cardiac myocytes increases. Certainly one can manipulate osmotic pressure to increase cell size and the authors have shown how this can be used to alter osteogenic differentiation of MSC in 3D alginate gels. However, is this simply a phenomenological observation or is there a stronger link to a physiological process?. The authors added a statement "Osteogenic differentiation of MSCs in stress relaxing microenvironments in 3D occurs during fracture healing in vivo^{11,12...}", but this is a little misleading as the stress relaxing microenvironment are simply the hydrogels that the Mooney group developed and not the fracture callous.

3. The authors should comment on how the time scales for their stress relaxation (~ 20 seconds, fast, and ~10 minutes, slow) leads to the long term changes in the MSC and differentiation (which occurs over a time scale of days to weeks). Does the stress relaxation matter or simply the local mechanical properties? Or are the local mechanical properties less important and it is simply about cell shape? While I understand that these are difficult to measure and deconvolute experimentally, It would be helpful if the authors could expand the discussion about their hypotheses.

4. Given that calcium signaling is involved, it does raise the issue as to whether or not the alginate chemistry itself is attributing to some of the observation. Would similar trends be observed in other stress relaxing gels that do not complex with calcium?

Response to Reviewer #1

Lee et al. have made a significant amount of effort addressing mine and other referees' concerns. I am fine with the volume measurements itself now with the added comparison with Zeiss super-resolution and detailed measurements of optical bias in different matrices. The added calcium imaging and TRPV4 clarification are also interesting and consistent with several previous literatures showing that TRPV4 and calcium signaling are responsible for osteogenesis, as the authors pointed out. However, I still don't think that the current experimental results presented in the revised manuscript are enough to claim that cell volume expansion is a critical factor here. Instead, it is very possible that TRPV4 activation is the key here, which induces both osteogenesis and volume change. I still think the experimental data itself is very interesting, however, I am not convinced by the current argument.

We are pleased that the reviewer is satisfied with the volume measurements, and thank the reviewer for a thorough reading of the manuscript and raising constructive critiques.

1. As the authors described in their response to my previous comment "calcium concentration in single cells was significantly different for cells in different relaxing hydrogels and under different osmotic pressure, which implicates changes in intracellular ion concentration as a major driver of cell volume difference", the observed volume change might be a consequence of the TRPV4 activation and ion concentration change, instead of a driving force of osteogenesis. The data certainly show that fast relaxing gel allows cell volume expansion, and results in TRPV4 activation. However, TRPV4 activation itself may induce both osteogenesis, as the authors pointed out, and cell volume expansion, also as the authors argue. In fact, the change in TRPV4 protein expression level and calcium amount are consistent with the level of osteogenesis under various conditions. So, the question becomes "is volume expansion necessary?" I don't see any current data isolating these effects and to show that volume change is required in regulating osteogenesis.

We thank the reviewer for raising this important point. In this revision, we have performed two additional experiments to directly test the influence of TRPV4 activation versus volume expansion. First, we cultured MSCs in slow relaxing gels, where volume expansion is restricted, and added the TRPV4 agonist (GSK101). We observed no increase in the proportion of ALP-positive cells upon TRPV4 activation. Further, we performed an experiment where volume expansion was restricted in fast relaxing gels by high hyper-osmotic pressure in the presence of the TRPV4 agonist. Again, we saw no increase in ALP-positive cells. Together these results indicate that activation of TRPV4 is not sufficient for driving osteogenesis, and that volume expansion is necessary for osteogenesis. These results are now included in Supplementary Figure 12 and we have commented on the findings in the text on page 8 line 184-186.

2. A possible role of volume expansion however could be mechanically triggering further relaxation of the matrix. As hypertonic pressure is applied, cells shrink to allow the matrix to relax and reorganize around the cell and confine them. Then it becomes similar to a slow relaxing matrix, from the perspective of cell confinement. If a hypotonic pressure is applied, cells expand to force the surrounding matrix to relax. Then this pressure essentially facilitates the gel to relax and allow cell spreading. I think matrix relaxation might actually be the key factor here.

We agree with the reviewer that matrix relaxation is a key factor, as we have shown that faster relaxing matrices promote volume expansion and TRPV4 activation, which lead to enhanced osteogenesis. To provide some insight into the point the reviewer makes about matrix remodeling, we have assessed the spatiotemporal properties of the matrix by evaluating the extents of polymer accumulation along the cell periphery using fluorescently-labeled alginate. No significant differences in the extent of alginate accumulation are observed for cells cultured in fast relaxing matrices with and without hyperosmotic pressure. However, this accumulation is significantly different between fast relaxing matrices under

hyperosmotic pressure and slow relaxing matrices, both conditions in which volume expansion and osteogenesis are inhibited. Thus, local remodeling per se does not control osteogenesis. These results are now included in Supplementary Figure 6 and are discussed in the discussion section of the manuscript (page 15, line 339-344).

3. The authors claim that calcium ions are responsible for changing the osmotic pressure balance and thus regulate cell volume. However, the calcium ion imaging is not quantitative. Calcium ions are indeed critical in cell physiology as they trigger many key functions such as muscle contraction, signaling, heart beating. However, it is not as concentrated as other ions such as Na and K in the cellular environment. How much osmotic pressure difference can be generated by Calcium channels moving Calcium ions along? The authors observed a volume change of 2 folds. This suggests that the amount of ions changed 2 fold in different matrices. It is unlikely to be dominated by Calcium ions given a low concentration is typically required for its functionality.

We thank the reviewer for this insightful comment. Unfortunately, due to the calcium imaging system and our gels, we are unable to quantitatively measure the absolute calcium concentration within the cells. We have clarified our description in the text to better convey that we have performed calcium imaging to demonstrate the activity of TRPV4 channels, and not to account for the total change in osmotic pressure (page 8, line 171-173). As the reviewer notes, many other ions in the cells could alter the osmotic balance.

Response to Reviewer #2

The revision is only partially responsive to previous comments, and the authors should also be more precise in their explanations (a couple of examples are given below but elsewhere also).

1. The authors need to write in the main text their response: "varying osmotic pressure was used as a method to independently modulate cell volume, and not to simulate conditions that might occur physiologically." Secondly, although they added in Introduction some words about in vivo osteogenesis being 2D, they must change the word "necessary" in describing 3D because even Ref.11 shows some osteogenesis in the absence of matrix remodeling: "matrix remodeling is necessary for osteogenic differentiation, and can occur through either protease-mediated degradation 10 or physical remodeling of matrices that are viscoelastic and exhibit fast stress relaxation 11"

We have revised the text indicated by the reviewer to better describe the findings in our manuscript and those previously reported by others. Notably, we have removed the claims that matrix remodeling is necessary for osteogenesis, as osteogenesis still occurs to some extent in the slow relaxing matrices in the referenced paper and in our manuscript.

2. The authors provide no data to address the question: How do the mechanical parameters of these gels vary in space and time around the cells? In their previous study (Chaudhuri et al., Nature Materials 2016), fast relaxing polymer accumulated around cells, which would imply a local stiffening. Does that occur in the present studies and when in relation to differentiation?

We thank the reviewer for the suggested experiment to assess the spatiotemporal properties of the matrix. We have evaluated the extent of polymer accumulation along the cell periphery using fluorescently-labeled alginate (Supplementary Figure 6). We do observe an increase in alginate accumulation around cells cultured in fast relaxing matrices. However, this accumulation is also found for cells in fast relaxing matrices under hyperosmotic pressure, where volume expansion and osteogenesis are not enhanced. Thus, we can conclude that alteration of the local matrix itself does not control osteogenesis. These results are now included in Supplementary Figure 6 and are discussed in the discussion section of the manuscript (page 15, line 339-344).

3. Questions about contractility and TRPV4 were poorly addressed. Contractility has long been shown to be pro-osteogenic in 2D (eg. Ref.1), and the authors now confirm this for their 3D (as should be stated in main text) together with cell volume measurements in new Fig.S13, but they did not address: "What is the effect of TRPV4 regulation ... on actomyosin organization and contractility." Given the new data showing (as expected) that contractility inhibitors suppress cell volume and osteogenesis, it is now critical for the authors to do another experiment that addresses whether their TRPV4 agonist (which enhanced osteogenic differentiation in hydrogels ... with an intermediate stress relaxation) over-rides the inhibitory effects of blebbistatin. This will address whether TRPV4 is simply upstream of contractility, which is a pathway that is established in the literature (per previous review comment). The authors must also explain in the main text the relevance of their results to osteogenesis in TRPV4 knockout mice such as those studied years ago in: "Impaired pressure sensation in mice lacking TRPV4. Suzuki M et al. J Biol Chem. 2003"

We thank the reviewer for suggesting an experiment to clarify the relationship between TRPV4 activity and contractility in controlling osteogenesis. As suggested, we have cultured cells in the presence of both blebbistatin and the TRPV4 agonist, and found a significant increase in ALP-positive cells compared to culture with blebbistatin alone (Supplementary Figure 17). The TRPV4 agonist restored the proportion of ALP-positive cells to similar levels as vehicle-treated controls. These results indicate that TRPV4 activity is not upstream of contractility.

We have also added a description of the prior publication (Suzuki *et al.*, *J Biol Chem.*, 2003) in the discussion section of the manuscript (page 13, line 293-294).

4. The added proliferation data for TRPV4 and the other conditions suggests proliferation is key to enhanced differentiation. This needs to be examined and discussed.

We have performed additional experiments to further elucidate the role of proliferation in regulating differentiation. We treated cells with mitomycin C to inhibit proliferation, and cultured them in medium relaxing gels with or without the TRPV4 agonist. No decrease in osteogenic differentiation was observed when proliferation was inhibited compared to the vehicle control (Supplementary Figure 13). Further, a significant increase in differentiation in the presence of the TRPV4 agonist was found, despite the inhibition of proliferation, suggesting that MSCs are still responsive to TRPV4 activation when proliferation is inhibited. Previously, we showed that higher cell concentration does not lead to enhanced differentiation either. Thus, while proliferation is highest in the groups that show greater levels of osteogenic differentiation, differentiation does not depend on proliferation.

Response to Reviewer #3

In this revision, the authors have been highly responsive to the original critiques, addressing many of the technical critiques of the experimental methods and analysis of the results. At this stage, some questions remain (detailed below), but I am less certain about the physiological relevance of the studies. The authors have attempted to address this in their revision, but it is not convincing. The results may likely be observations that occur when MSCs are cultured in a narrow subset of hydrogels used by bioengineers (alginates that also bind calcium) and that the findings may be less applicable to any observations of MSCs in vivo. While the reviewer values that MSCs are used for many regenerative medicine applications and these results could assist in the design of biomaterials scaffolds for them, it is less clear that this will be of broader interest and it may be better suited for a more specialized tissue engineering or biomaterials journal audience. However, I trust the editor to make this final decision.

We appreciate that the reviewer found our first revision to be highly responsive, with the original critiques largely addressed. We thank the reviewer for the additional constructive critiques. As we argue below in our response to comment 2, there is evidence that these results are relevant *in vivo*, in addition to being relevant for applications in regenerative medicine and the design of biomaterial scaffolds.

1. The MSCs were cultured in osteogenic and adipogenic induction media (mixed media), so my comment remains that the MSC characterization is weak. Certainly key markers of osteogenic differentiation are reported, but this should be strengthened and multiple markers at the gene and protein level reported. Also, given the media used, adipogenic markers should be reported.

We have performed qRT-PCR to assess gene expression of four additional markers of osteogenic differentiation: osteocalcin, RUNX2, alkaline phosphatase, and bone sialoprotein. We found a significant decrease in all of these markers when cell volume expansion was inhibited with hyperosmotic pressure in a dose-dependent manner (Supplementary Figure 8). Additionally, we evaluated gene expression of three adipogenic markers: CD36, adiponectin, and fatty acid binding protein 4. We did not find any significant differences in these markers under different levels of hyper-osmotic pressure. This result is in agreement with our prior finding that the stemness marker CD105 increases with increasing osmotic pressure. Thus, MSCs maintain their stemness when volume expansion is restricted.

2. I have been unable to find any reports in the literature where such a dramatic volume change occurs in MSCs in vivo. What is the physiological relevance of the volume change? Answering this question would help improve the impact of this work. For example, during cardiac hypertrophy and disease progression, the size of the cardiac myocytes increases. Certainly one can manipulate osmotic pressure to increase cell size and the authors have shown how this can be used to alter osteogenic differentiation of MSC in 3D alginate gels. However, is this simply a phenomenological observation or is there a stronger link to a physiological process?_The authors added a statement "Osteogenic differentiation of MSCs in stress relaxing microenvironments in 3D occurs during fracture healing in vivo^{11,12...}", but this is a little misleading as the stress relaxing microenvironment are simply the hydrogels that the Mooney group developed and not the fracture callous.

The reviewer brings up an important point about physiological relevance. We too have been unable to find reports of cell volume changes in MSCs *in vivo*. However, the two manuscripts cited from the Mooney group in the statement noted above do include measurements of stress relaxation of human fracture hematomas (Fig. 1a in ref. 11, Fig. 1a in ref. 12). These were shown to be similar to the relaxation rate of our fast relaxing gels. Additionally, McDonald and colleagues reported a very similar rate for rat fracture calluses (Table. 1 in McDonald, et al., *Journal of Orthopaedic Research*, 2009). As MSCs play a key role in fracture healing and are found within the fracture hematoma (Bielby, et al.,

Injury, 2007 and Dimitriou, et al., *Injury*, 2005), it is reasonable to conclude that the impact of fast stress relaxation on MSCs is likely to be physiologically relevant.

3. The authors should comment on how the time scales for their stress relaxation (~ 20 seconds, fast, and ~10 minutes, slow) leads to the long-term changes in the MSC and differentiation (which occurs over a time scale of days to weeks). Does the stress relaxation matter or simply the local mechanical properties? Or are the local mechanical properties less important and it is simply about cell shape? While I understand that these are difficult to measure and deconvolute experimentally, it would be helpful if the authors could expand the discussion about their hypotheses.

We have added text in the discussion of the time scales of stress relaxation to emphasize that the time scales reported are simply one metric used to compare viscoelasticity (the time it takes to reach 50% of the initial stress under the specific conditions of our rheological experiment). We note that cell spreading and expansion are dynamic processes in which cells are continuously applying strains to the matrix, so that stress relaxation is continuously occurring. Additionally, as discussed in response to the first reviewer, we have provided additional data showing local matrix remodeling (accumulation of fluorescently-labeled alginate) in fast relaxing gels. We also demonstrate that local remodeling occurs under hyperosmotic pressure, when osteogenic differentiation is greatly reduced, and thus local remodeling per se does not explain the results we present in this manuscript. We discuss these results in the discussion section of the manuscript (page 15, line 339-344).

4. Given that calcium signaling is involved, it does raise the issue as to whether or not the alginate chemistry itself is attributing to some of the observation. Would similar trends be observed in other stress relaxing gels that do not complex with calcium?

We thank the reviewer for raising this important point. We have plotted the proportion of ALP-positive MSCs vs. calcium concentration in the gels for a number of conditions in this manuscript. There is no significant correlation between differentiation and calcium concentration (Supplementary Figure 9). Thus, we are confident that the observed phenomena are not simply a response to calcium crosslinking of the alginate.

Reviewers' Comments:

Reviewer #1:

None

Reviewer #2:

Remarks to the Author:

The authors were reasonably responsive to previous concerns.

New Fig.S17 is surprising and novel in showing osteo-suppression that results from inhibiting blebbistatin can be rescued by GSK101. However, Fig.7a needs to include the volume measurements for both the single and dual drug treatments in this experiment. Furthermore, the specificity of GSK101 needs to be verified in this assay with GSK101 + blebbistatin + knockdown of TRPV4 (siRNA, shRNA, CRISPR...). If GSK101 is specific, then this experiment should show that ALP is reduced relative to a knockdown control.

Secondly, the authors need to revise their Discussion about knockout mice and write "...TRPV4-null mice have no reported defects in bone (41)."

Response to Reviewer #2

The authors were reasonably responsive to previous concerns.

We are pleased that the reviewer found our previously revised manuscript to be reasonably responsive.

New Fig.S17 is surprising and novel in showing osteo-suppression that results from inhibiting blebbistatin can be rescued by GSK101. However, Fig.7a needs to include the volume measurements for both the single and dual drug treatments in this experiment. Furthermore, the specificity of GSK101 needs to be verified in this assay with GSK101 + blebbistatin + knockdown of TRPV4 (siRNA, shRNA, CRISPR...). If GSK101 is specific, then this experiment should show that ALP is reduced relative to a knockdown control.

We thank the reviewer for the additional comments. As suggested, we have measured cell volumes for both the single and dual drug treatments. This result is now included in Supplementary Figure 17b and Figure 7a. Regarding the specificity of GSK101, many previous studies have already shown that GSK 101 specifically activates TRPV4. These include the following:

1. Thorneloe, Kevin S., et al. "N-((1S)-1-{{4-((2S)-2-{{(2, 4-dichlorophenyl) sulfonyl} amino}-3-hydroxypropanoyl)-1-piperazinyl}carbonyl}-3-methylbutyl)-1-benzothiophene-2-carboxamide (GSK1016790A), a novel and potent transient receptor potential vanilloid 4 channel agonist induces urinary bladder contraction and hyperactivity: Part I." *Journal of Pharmacology and Experimental Therapeutics* 326.2 (2008): 432-442.

The authors applied GSK101 into the TRPV4^{+/+} mice and the TRPV4^{-/-} mice in investigating the role of TRPV4 in the urinary bladder. The results showed that GSK101 specifically activates TRPV4 channels.

2. Mendoza, Suelhem A., et al. "TRPV4-mediated endothelial Ca²⁺ influx and vasodilation in response to shear stress." *American Journal of Physiology-Heart and Circulatory Physiology* 298.2 (2009): H466-H476.

The authors applied GSK101 and showed increased endothelial Ca²⁺ and induced potent relaxation of small mesenteric arteries from wild-type (WT) but not TRPV4^{-/-} mice. This result supported that GSK101 specifically activates TRPV4 channels.

3. Jin, Min, et al. "Determinants of TRPV4 activity following selective activation by small molecule agonist GSK1016790A." *PLoS One* 6.2 (2011): e16713.

This work showed that GSK101 specifically activates TRPV4 channels, leading to a rapid partial desensitization and downregulation of the channel expression on the plasma membrane.

4. Mihara, Hiroshi, et al. "Transient receptor potential vanilloid 4 (TRPV4)-dependent calcium influx and ATP release in mouse oesophageal keratinocytes." *The Journal of physiology* 589.14 (2011): 3471-3482.

The authors applied GSK101 to WT keratinocytes and TRPV4 knockout keratinocytes and observed GSK101 increased cytosolic Ca²⁺ concentrations in cultured WT keratinocytes but not in TRPV4 knockout (KO) cells. This result provides additional evidence that GSK101 specifically activates TRPV4 channels.

Secondly, the authors need to revise their Discussion about knockout mice and write "...TRPV4-null mice have no reported defects in bone (41)."

We have now revised the text to include this exact wording.